# Stability and Generalization of Bilevel Programming in Hyperparameter Optimization

**Fan Bao**\*, **Guoqiang Wu**\*, **Chongxuan Li**\*, **Jun Zhu**†, **Bo Zhang**
Dept. of Comp. Sci. & Tech., Institute for AI, Tsinghua-Huawei Joint Center for AI
BNRist Center, State Key Lab for Intell. Tech. & Sys., Tsinghua University, Beijing, China
bf19@mails.tsinghua.edu.cn,{guoqiangwu90, chongxuanli1991}@gmail.com,
{dcszj, dcszb}@tsinghua.edu.cn

## Abstract

The (gradient-based) bilevel programming framework is widely used in hyperparameter optimization and has achieved excellent performance empirically. Previous theoretical work mainly focuses on its optimization properties, while leaving the analysis on generalization largely open. This paper attempts to address the issue by presenting an expectation bound w.r.t. the validation set based on uniform stability. Our results can explain some mysterious behaviours of the bilevel programming in practice, for instance, overfitting to the validation set. We also present an expectation bound for the classical cross-validation algorithm. Our results suggest that gradient-based algorithms can be better than cross-validation under certain conditions in a theoretical perspective. Furthermore, we prove that regularization terms in both the outer and inner levels can relieve the overfitting problem in gradient-based algorithms. In experiments on feature learning and data reweighting for noisy labels, we corroborate our theoretical findings.

## 1 Introduction

Hyperparameter optimization (HO) is a common problem arising from various fields including neural architecture search [24, 8], feature learning [10], data reweighting for imbalanced or noisy samples [33, 37, 9, 35], and semi-supervised learning [14]. Formally, HO seeks the hyperparameter-hypothesis pair that achieves the lowest expected risk on testing samples from an unknown distribution. Before testing, however, only a set of training samples and a set of validation ones are given. The validation and testing samples are assumed to be from the same distribution. In contrast, depending on the task, the underlying distributions of the training and testing samples can be the same [24, 10] or different [33].

Though various methods [17] have been developed to solve HO (See Section 5 for a comprehensive review), the *bilevel programming* (BP) [6, 32, 10] framework is a natural solution and has achieved excellent performance in practice [24]. BP consists of two nested search problems: in the inner level, it seeks the best hypothesis (e.g., a prediction model) on the training set given a specific configuration of the hyperparameter (e.g., the model architectures), while in the outer level, it seeks the hyperparameter (and its associated hypothesis) that results in the best hypothesis in terms of the error on the validation set.

---

\*Equal contribution. G. Wu is now at School of Software, Shandong University and C. Li is now at Gaoling School of AI, Renmin University of China. The work was done when they were at Tsinghua University.
†Corresponding author.

35th Conference on Neural Information Processing Systems (NeurIPS 2021).

Generally, it is hard to solve the BP problem exactly, and several approximate algorithms have been developed in previous work. As a classical approach, *Cross-validation* (CV)[3] can be viewed as an approximation method. It obtains a finite set of hyperparameters via grid search [31] or random search [2] as well as a set of the corresponding hypothesises trained in the inner level, and selects the best hyperparameter-hypothesis pair according to the validation error. CV is well understood in theory [31, 36] but suffers from the issue of scalability [10]. Recently, alternative algorithms [9, 10, 11, 29, 35] based on *unrolled differentiation* (UD) have shown promise on tuning up to millions of hyperparameters. In contrast to CV, UD exploits the validation data more aggressively: it optimizes the validation error directly via (stochastic) gradient descent in the space of the hyperparameters, and the gradient is obtained by finite steps of unrolling in the inner level.

Though promising, previous theoretical work [10, 39, 35] of UD mainly focuses on the optimization, while leaving its generalization analysis largely open. This paper takes a first step towards solving it and aims to answer the following questions rigorously:

- *Can we obtain certain learning guarantees for UD and insights to improve it?*
- *When should we prefer UD over classical approaches like CV in a theoretical perspective?*

Our first main contribution is to present a notion of *uniformly stable on validation (in expectation)* and an expectation bound of UD algorithms (with stochastic gradient descent in the outer level) on the validation data as follows:

$$|\text{Expected risk of UD} - \text{Empirical risk of UD on validation}| \lesssim \tilde{\mathcal{O}}\left(\frac{T^\kappa}{m}\right) \text{ and } \tilde{\mathcal{O}}\left(\frac{(1+\eta\gamma_\varphi)^{2K}}{m}\right),$$
(1)

where $T$ and $K$ are the numbers of steps in the outer level and the inner level respectively, $\kappa \in (0,1)$ is a constant, $\eta$ is the learning rate in the inner level, $\gamma_\varphi$ is a smoothness coefficient of the inner loss, $m$ is the size of the validation set and $\lesssim$ means the inequality holds in expectation. As detailed in Section 4.1, our results not only present the order of the important factors in the generalization gap but also explain some mysterious behaviours of UD in practice, for instance, the trade-off on the values of $T$ and $K$ [10].

Our second main contribution is to systematically compare UD and CV from the perspective of generalization in Section 4.2. Instead of using the existing high probability bounds of CV (see Theorem 4.4 in [31]), we present an expectation bound of CV for a direct comparison to Eq. (1) as follows:[4]

$$|\text{Expected risk of CV} - \text{Empirical risk of CV on validation}| \lesssim \mathcal{O}\left(\sqrt{\frac{\log T}{m}}\right).$$
(2)

On the one hand, Eq. (1) and Eq. (2) suggest that with a large $T$,[5] CV has a much lower risk of overfitting than UD. On the other hand, the dependence of Eq. (2) on $m$ is $\mathcal{O}\left(\sqrt{\frac{1}{m}}\right)$, which is worse than that of UD. Furthermore, as discussed before, probably UD has a much lower validation risk than CV. Indeed, we show that CV with random search suffers from the curse of dimensionality. These analyses may explain the superior performance of UD [24], especially when we have a reasonable choice of $T$, a sufficiently large $m$ and a sufficiently high-dimensional hyperparameter space.

Our third main contribution is to present a regularized UD algorithm in Section 4.3 and prove that both a weight decay term of the parameter in the inner level and that of the hyperparameter in the outer level can increase the stability of the UD algorithms. Thus, the generalization performance will probably improve if the terms do not hurt the validation risk too much.

Finally, experiments presented in Section 6 validate our theory findings. In particular, we reproduce the mysterious behaviours of UD (e.g., overfitting to the validation data) observed in [10], which can be explained via Eq. (1). Besides, we empirically compare UD and CV and analyze their performance via Eq. (1) and Eq. (2). Further, we show the promise of the regularization terms in both levels.

---

[3]While CV can refer to a general class of approaches by splitting a training set and a validation set, we focus on grid search and random search and denote the two methods as CV collectively.

[4]We do not find a strong dependency of CV on $K$ in both the theory and practice.

[5]Every hyperparameter considered corresponds to a loop for the inner level optimization, which is the computational bottleneck in HO. Therefore, for fairness, we assume UD and CV share the same $T$ by default.

## 2 Problem Formulation

Let $Z$, $\Theta$ and $\Lambda$ denote the data space, the hypothesis space and the hyperparameter space respectively. $\ell : \Lambda \times \Theta \times Z \to [a, b]$ is a bounded loss function[6] and its range is $s(\ell) = b - a$.

Given a hyperparameter $\lambda \in \Lambda$, a hypothesis $\theta \in \Theta$ and a distribution $D$ on the data space $Z$, $R(\lambda, \theta, D)$ denotes the expected risk of $\lambda, \theta$ on $D$, i.e., $R(\lambda, \theta, D) = \mathbb{E}_{z \sim D}[\ell(\lambda, \theta, z)]$. Hyperparameter optimization (HO) seeks the hyperparameter-hypothesis pair that achieves the lowest expected risk on testing samples from an unknown distribution.

Before testing, however, only a training set $S^{tr}$ of size $n$ and a validation set $S^{val}$ of size $m$ are accessible. As mentioned in Section 1, we consider a general HO problem, where the distribution of the testing samples $D^{te}$ is assumed to be the same as that of the validation ones $D^{val}$ but can differ from that of the training ones $D^{tr}$. In short, we assume $D^{val} = D^{te}$, but we **do not** require $D^{tr} = D^{te}$. Given a hyperparameter $\lambda \in \Lambda$, a hypothesis $\theta \in \Theta$ and the validation set $S^{val}$, $\hat{R}^{val}(\lambda, \theta, S^{val})$ denotes the empirical risk of $\lambda, \theta$ on $S^{val}$, i.e., $\hat{R}^{val}(\lambda, \theta, S^{val}) = \frac{1}{m} \sum_{i=1}^{m} \ell(\lambda, \theta, z_i^{val})$. The empirical risk on training is defined as $\hat{R}^{tr}(\lambda, \theta, S^{tr}) = \frac{1}{n} \sum_{i=1}^{n} \varphi_i(\lambda, \theta, z_i^{tr})$, where $\varphi_i$ can be a slightly modified (e.g., reweighted) version of $\ell$ for $i$-th training sample[7].

Technically, an HO algorithm $\mathbf{A}$ is a function mapping $S^{tr}$ and $S^{val}$ to a hyperparameter-hypothesis pair, i.e., $\mathbf{A} : Z^n \times Z^m \to \Lambda \times \Theta$. In contrast, a *randomized* HO algorithm does not necessarily return a deterministic hyperparameter-hypothesis pair but more generally a random variable with the outcome space $\Lambda \times \Theta$.

Though extensive methods [17] have been developed to solve HO (See Section 5 for a comprehensive review), the *bilevel programming* [6, 32, 10] is a natural solution and has achieved excellent performance in practice recently [24]. It consists of two nested search problems as follows:

$$\underbrace{\lambda^*(S^{tr}, S^{val}) = \arg\min_{\lambda \in \Lambda} \hat{R}^{val}(\lambda, \theta^*(\lambda, S^{tr}), S^{val})}_{\text{Outer level optimization}}, \text{ where } \underbrace{\theta^*(\lambda, S^{tr}) = \arg\min_{\theta \in \Theta} \hat{R}^{tr}(\lambda, \theta, S^{tr})}_{\text{Inner level optimization}}.$$

(3)

In the *inner level*, it seeks the best hypothesis on $S^{tr}$ given a specific configuration of the hyperparameter. In the *outer level*, it seeks the hyperparameter $\lambda^*(S^{tr}, S^{val})$ (and its associated hypothesis $\theta^*(\lambda^*(S^{tr}, S^{val}), S^{tr})$) that results in the best hypothesis in terms of the error on $S^{val}$. Eq. (3) is sufficiently general to include a large portion of the HO problems we are aware of, such as:

- **Differential Architecture Search [24]** Let $\lambda$ be the coefficients of a set of network architecture, let $\theta$ be the parameters in the neural network defined by the coefficients, and let $l$ be the cross-entropy loss associated with $\theta$ and $\lambda$. Namely, $l(\lambda, \theta, z) = CE(h_{\lambda,\theta}(x), y)$, where $h_{\lambda,\theta}(\cdot)$ is a neural network with architecture $\lambda$ and parameter $\theta$.

- **Feature Learning [10]** Let $\lambda$ be a feature extractor, let $\theta$ be the parameters in a classifier that takes features as input and let $l$ be the cross-entropy loss associated with $\theta$ and $\lambda$. Namely, $l(\lambda, \theta, z) = CE(g_\theta(h_\lambda(x)), y)$, where $h_\lambda(\cdot)$ is the feature extractor and $g_\theta(\cdot)$ is the classifier.

- **Data Reweighting for Imbalanced or Noisy Samples [35]** Let $\lambda$ be the coefficients of the training data, let $\theta$ be the parameters of a classifier that takes the data as input, let $l$ be the cross-entropy loss associated with $\theta$. Namely, $l(\theta, z) = CE(h_\theta(x), y)$, where $h_\theta(\cdot)$ is the classifier and $l$ is irrelevant to $\lambda$. The empirical risk on the training set is defined through a *reweighted* version of the loss $\varphi_i(\lambda, \theta, z_i) = \sigma(\lambda_i) l(\theta, z_i)$, where $\sigma(\cdot)$ is the sigmoid function and $\lambda_i$ is the coefficient corresponding to $z_i$.

---

[6]This formulation also includes the case where the loss function is irrelevant to the hyperparameter. Also see Appendix F for a discussion of the boundedness assumption of the loss function.

[7]While in many tasks such as differential architecture search [24] and feature learning [10], $\varphi_i$ is just the same as $\ell$, we distinguish between them to include tasks where $\varphi_i$ and $\ell$ are different, such as data reweighting [35].

**Algorithm 1** Unrolled differentiation for hyperparameter optimization

1: **Input:** Number of steps $T$ and $K$; initialization $\hat{\theta}_0$ and $\hat{\lambda}_0$; learning rate scheme $\alpha$ and $\eta$
2: **Output:** The hyperparameter $\hat{\lambda}_{ud}$ and hypothesis $\hat{\theta}_{ud}$
3: **for** $t = 0$ **to** $T - 1$ **do**
4:      $\hat{\theta}_0^t \leftarrow \hat{\theta}_0$
5:      **for** $k = 0$ **to** $K - 1$ **do**
6:          $\hat{\theta}_{k+1}^t \leftarrow \hat{\theta}_k^t - \eta_{k+1} \nabla_\theta \hat{R}^{tr}(\hat{\lambda}_t, \theta, S^{tr})|_{\theta = \hat{\theta}_k^t}$
7:      **end for**
8:      $\hat{\lambda}_{t+1} \leftarrow \hat{\lambda}_t - \alpha_{t+1} \nabla_\lambda \hat{R}^{val}(\lambda, \hat{\theta}_K^t(\lambda), S^{val})|_{\lambda = \hat{\lambda}_t}$
9: **end for**
10: **return** $\hat{\lambda}_T$ and $\hat{\theta}_K^T$

**Algorithm 2** Cross-validation for hyperparameter optimization

1: **Input:** Number of steps $T$ and $K$; initialization $\{\hat{\lambda}_t\}_{t=1}^T$ and $\{\hat{\theta}_0^t\}_{t=1}^T$; learning rate scheme $\eta$
2: **Output:** The hyperparameter $\hat{\lambda}_{cv}$ and hypothesis $\hat{\theta}_{cv}$
3: **for** $k = 0$ **to** $K - 1$ **do**
4:      $\hat{\theta}_{k+1}^t \leftarrow \hat{\theta}_k^t - \eta_{k+1} \nabla_\theta \hat{R}^{tr}(\hat{\lambda}_t, \theta, S^{tr})|_{\theta = \hat{\theta}_k^t}$
5: **end for**
6: $t^* \leftarrow \underset{1 \leq t \leq T}{\arg\min} \hat{R}^{val}(\hat{\lambda}_t, \hat{\theta}_K^t, S^{val})$
7: **return** $\hat{\lambda}_{t^*}$ and $\hat{\theta}_K^{t^*}$

## 3 Approximate the Bilevel Programming Problem

In most of the situations (e.g., neural network as the hypothesis class [24]), the global optima of both the inner and outer level problems in Eq. (3) are nontrivial to achieve. It is often the case to approximate them in a certain way (e.g., using (stochastic) gradient descent) as follows:

$$\underbrace{\hat{\lambda}(S^{tr}, S^{val}) \approx \underset{\lambda \in \Lambda}{\arg\min}\, \hat{R}^{val}(\lambda, \hat{\theta}(\lambda, S^{tr}), S^{val})}_{\text{Approximate outer level optimization}}, \text{ where } \underbrace{\hat{\theta}(\lambda, S^{tr}) \approx \underset{\theta \in \Theta}{\arg\min}\, \hat{R}^{tr}(\lambda, \theta, S^{tr})}_{\text{Approximate inner level optimization}}.$$
(4)

Here $\hat{\theta}(\lambda, S^{tr})$ can be deterministic or random. In this perspective, we can view the unrolled differentiation (UD) and cross-validation (CV) algorithms as two implementation of Eq. (4).

**Unrolled differentiation.** The UD-based algorithms [9, 10, 11, 29, 35] solve Eq. (4) via performing finite steps of gradient descent in both levels. Given a hyperparameter, the inner level performs $K$ updates, and keeps the whole computation graph. The computation graph is a composite of $K$ parameter updating functions, which are differentiable with respect to $\lambda$, and thereby the memory complexity is $\mathcal{O}(K)$. As a result, the corresponding hypothesis is a function of the hyperparameter. The outer level updates the hyperparameter a single step by differentiating through inner updates, which has a $\mathcal{O}(K)$ time complexity, and the inner level optimization repeats given the updated hyperparameter. Totally, the outer level updates $T$ times. Formally, it is given by Algorithm 1, where we omit the dependency of $\lambda$ and $\theta$ on $S^{val}$ and $S^{tr}$ for simplicity.

In some applications [24, 10], $n$ and $m$ can be very large and we cannot efficiently calculate the gradients in Algorithm 1. In this case, stochastic gradient descent (SGD) can be used to update the hyperparameter (corresponding to line 8 in Algorithm 1) as follows:

$$\hat{\lambda}_{t+1} \leftarrow \hat{\lambda}_t - \alpha_{t+1} \nabla_\lambda \hat{R}^{val}(\lambda, \hat{\theta}_K^t(\lambda), \{z_j\})|_{\lambda = \hat{\lambda}_t},$$
(5)

where $z_j$ is randomly selected from $S^{val}$. Similarly, we can also adopt SGD when updating the hypothesis and then all intermediate hypothesises are random functions of $\lambda$ and $S^{tr}$.

**Cross-validation.** CV is a classical approach for HO. It first obtains a finite set of hyperparameters, which is often a subset of $\Lambda$, via grid search [31] or random search [2] [8]. Then, it separately trains the inner level to obtain the corresponding hypothesis given a hyperparameter. Finally, it selects the best hyperparameter-hypothesis pair according to the validation error. It is formally given by Algorithm 2, where we use gradient descent to approximate the inner level (i.e., line 4). We can also adopt SGD to update the hypothesis.

---

[8] In our experiments, the hyperparameter is too high-dimensional to perform grid search, and thus random search is preferable. Nevertheless, they will be shown to have similar theoretical properties.

In terms of optimization, CV searches over a prefixed subset of $\Lambda$ in a discrete manner, while UD leverages the local information of the optimization landscape (i.e., gradient). Therefore, UD is more likely to achieve a lower empirical risk on the validation data with the same $T$. In the following, we will discuss the two algorithms from the perspective of generalization.

## 4 Main Results

We present the main results below for clarity, and the readers can refer to Appendix A for all proofs.

### 4.1 Stability and Generalization of UD

In most of the recent HO applications [10, 24, 14, 33] that we are aware of, stochastic gradient descent (SGD) is adopted for its scalability and efficiency. Therefore, we present the main results on UD with SGD in the outer level here.[9]

Recall that a randomized HO algorithm returns a random variable with the outcome space $\Lambda \times \Theta$ in general. To establish the generalization bound, we define the following notion of *uniform stability on validation in expectation*.

**Definition 1.** *A randomized HO algorithm* $\mathbf{A}$ *is* $\beta$-*uniformly stable on validation in expectation if for all validation datasets* $S^{val}, S'^{val} \in Z^m$ *such that* $S^{val}, S'^{val}$ *differ in at most one sample, we have*

$$\forall S^{tr} \in Z^n, \forall z \in Z, \mathbb{E}_{\mathbf{A}} \left[ \ell(\mathbf{A}(S^{tr}, S^{val}), z) - \ell(\mathbf{A}(S^{tr}, S'^{val}), z) \right] \leq \beta.$$

Compared to existing work [15], Definition 1 considers the expected influence of changing one validation sample on a randomized HO algorithm. The reasons are two-folded. On the one hand, in HO, the distribution of the testing samples is assumed to be the same as that of the validation ones but can differ from that of the training ones [33], making it necessary to consider the bounds on the validation set. On the other hand, classical results in CV suggest that such bounds are usually tighter than the ones on the training set [31].

If a randomized HO algorithm is $\beta$-uniformly stable on validation in expectation, then we have the following generalization bound.

**Theorem 1** (Generalization bound of a uniformly stable algorithm)**.** *Suppose a randomized HO algorithm* $\mathbf{A}$ *is* $\beta$-*uniformly stable on validation in expectation, then*

$$|\mathbb{E}_{\mathbf{A}, S^{tr} \sim (D^{tr})^n, S^{val} \sim (D^{val})^m} \left[ R(\mathbf{A}(S^{tr}, S^{val}), D^{val}) - \hat{R}^{val}(\mathbf{A}(S^{tr}, S^{val}), S^{val}) \right]| \leq \beta.$$

Note that this is an expectation bound for any randomized HO algorithm with uniform stability. We now analyze the stability, namely bounding $\beta$, for UD with SGD in the outer level. Indeed, we consider a general family of algorithms that solve Eq. (4) via SGD in the outer level without any restriction on the inner level optimization in the following Theorem 2.

**Theorem 2** (Uniform stability of algorithms with SGD in the outer level)**.** *Suppose* $\hat{\theta}$ *is a random function in a function space* $\mathcal{G}_{\hat{\theta}}$ *and* $\forall S^{tr} \in Z^n$, $\forall z \in Z$, $\forall g \in \mathcal{G}_{\hat{\theta}}$, $\ell(\lambda, g(\lambda, S^{tr}), z)$ *as a function of* $\lambda$ *is* $L$-*Lipschitz continuous and* $\gamma$-*Lipschitz smooth, let* $c \leq \frac{s(\ell)}{2L^2}$ *and* $\kappa = \frac{c((1-1/m)\gamma)}{c((1-1/m)\gamma)+1}$. *Then, solving Eq. (4) with* $T$ *steps SGD and learning rate* $\alpha_t \leq \frac{c}{t}$ *in the outer level is* $\beta$-*uniformly stable on validation in expectation with*

$$\beta = \frac{2cL^2}{m} \left( \frac{1}{\kappa} \left( \left( \frac{Ts(\ell)}{2cL^2} \right)^{\kappa} - 1 \right) + 1 \right),$$

*which is increasing w.r.t.* $L$ *and* $\gamma$. *(Recall that* $s(\ell) = b - a$ *is the range of the loss.)*

Theorem 2 doesn't assume a specific form of $\hat{\theta}$. Indeed, UD instantiates $\hat{\theta}$ as the output of SGD or GD in the inner level. For SGD, the corresponding $\mathcal{G}_{\hat{\theta}}$ is formed by iterating over all possible random indexes and initializations in $\hat{\theta}$. For GD, the corresponding $\mathcal{G}_{\hat{\theta}}$ is only formed by iterating over all

---

[9]See Appendix D for the results on UD with GD in the outer level. The generalization gap of UD with GD in the outer level has an exponential dependence on $T$ and $K$, and has the same $1/m$ dependence on $m$ as SGD.

possible initializations in $\hat{\theta}$, since GD uses full batches. We analyze the constants $L$ and $\gamma$ appearing in $\beta$ of UD, which solves the inner level problem by either SGD or GD, given the following mild assumptions on the outer loss $\ell$ and the inner loss $\varphi_i$.

**Assumption 1.** $\Lambda$ *and* $\Theta$ *are compact and convex with non-empty interiors, and* $Z$ *is compact.*

**Assumption 2.** $\ell(\lambda, \theta, z) \in C^2(\Omega)$, *where* $\Omega$ *is an open set including* $\Lambda \times \Theta \times Z$ *(i.e.,* $\ell$ *is second order continuously differentiable on* $\Omega$*).*

**Assumption 3.** $\varphi_i(\lambda, \theta, z) \in C^3(\Omega)$, *where* $\Omega$ *is an open set including* $\Lambda \times \Theta \times Z$ *(i.e.,* $\varphi_i$ *is third order continuously differentiable on* $\Omega$*).*

**Assumption 4.** $\varphi_i(\lambda, \theta, z)$ *is* $\gamma_\varphi$*-Lipschitz smooth as a function of* $\theta$ *for all* $1 \leq i \leq n$, $z \in Z$ *and* $\lambda \in \Lambda$ *(Assumption 1 and Assumption 3 imply such a constant* $\gamma_\varphi$ *exists).*

**Theorem 3.** *Suppose Assumption 1,2,3,4 hold and the inner level problem is solved with* $K$ *steps SGD or GD with learning rate* $\eta$, *then* $\forall S^{tr} \in Z^n$, $\forall z \in Z$, $\forall g \in \mathcal{G}_{\hat{\theta}}$, $\ell(\lambda, g(\lambda, S^{tr}), z)$ *as a function of* $\lambda$ *is* $L = \mathcal{O}((1 + \eta\gamma_\varphi)^K)$ *Lipschitz continuous and* $\gamma = \mathcal{O}((1 + \eta\gamma_\varphi)^{2K})$ *Lipschitz smooth.*

**Remark:** Generally, a neural network composed of smooth operators satisfy all assumptions. We notice that the continuously differentiable assumption in Assumption 2 and Assumption 3 does not hold for ReLU. However, we argue that there are many smooth approximations of ReLU including Softplus, Gelu [16], and Lipswish [5], which satisfy the assumption and achieve promising results in classification and deep generative modeling.

Combining the results in Theorem 1, Theorem 2 and Theorem 3, we obtain an expectation bound of UD that depends on the number of steps in the outer level $T$, the number of steps in the inner level $K$ and the validation sample size $m$. Roughly speaking, its generalization gap has an order of $\tilde{\mathcal{O}}(\frac{T^\kappa}{m})$ or $\tilde{\mathcal{O}}(\frac{(1+\eta\gamma_\varphi)^{2K}}{m})$. In Appendix B, we construct a worst case where the Lipschitz constant $L$ in Theorem 3 increases at least exponentially w.r.t. $K$. According to Theorem 2, the stability bound also increases exponentially w.r.t. $K$ in the worst case. Besides, if we further assume the inner loss $\varphi_i$ is convex or strongly convex, we can derive tighter generalization gaps. Indeed, the dependence on $K$ of the generalization gap is $\mathcal{O}(K^2)$ in the convex case and $\mathcal{O}(1)$ in the strongly convex case. Please see Appendix C for a complete proof.

Our results can explain some mysterious behaviours of the UD algorithms in practice [10]. According to Theorem 2 and Theorem 3, very large values of $K$ and $T$ will significantly decrease the stability of UD (i.e., increasing $\beta$), which suggests a high risk of overfitting. On the other hand, if we use very small $T$ and $K$, the empirical risk on the validation data might be insufficiently optimized, probably leading to underfitting. This trade-off on the values of $K$ and $T$ has been observed in previous theoretical work [10], which mainly focuses on optimization and does not provide a formal explanation. We also confirmed this phenomenon in two different experiments (See results in Section 6.2).

As for the number of validation data $m$, the generalization gap has an order of $\mathcal{O}(\frac{1}{m})$, which is satisfactory compared to that of CV as presented in Section 4.2.

## 4.2 Comparison with CV

CV is a classical approach for HO with theoretical guarantees, which serves as a natural baseline of our results on UD in Theorem 1, Theorem 2 and Theorem 3. However, existing results on CV (see Theorem 4.4 in [31]) are in the form of high probability bounds, which are not directly comparable to ours. To compare under the same theoretical framework to obtain meaningful conclusions, we present an expectation bound for CV as follows.

**Theorem 4** (Expectation bound of CV). *Suppose* $S^{tr} \sim (D^{tr})^n$, $S^{val} \sim (D^{val})^m$ *and* $S^{tr}$ *and* $S^{val}$ *are independent, and let* $\mathbf{A}^{cv}(S^{tr}, S^{val})$ *denote the results of CV as shown in Algorithm 2, then*

$$|\mathbb{E}\left[R(\mathbf{A}^{cv}(S^{tr}, S^{val}), D^{val}) - \hat{R}^{val}(\mathbf{A}^{cv}(S^{tr}, S^{val}), S^{val})\right]| \leq s(\ell)\sqrt{\frac{\log T}{2m}}.$$

Technically, the expectation bound of CV is proved via the property of the maximum of a set of subgaussian random variables (see Theorem 1.14 in [34]), which is distinct from the union bound used in the high probability bound of CV (see Theorem 4.4 in [31]). In Appendix G.2, we also verify the $\mathcal{O}(\sqrt{\frac{1}{m}})$ dependence of the expectation bound empirically.

On one hand, we note that the growth of the generalization gap w.r.t. $T$ is logarithmic in Theorem 4, which is much slower than that of our results on UD. Besides, it does not explicitly depend on $K$. Therefore, sharing the same large values of $K$ and $T$, CV has a much lower risk of overfitting than UD. On the other hand, the dependence of Theorem 4 on $m$ is $\mathcal{O}(\sqrt{\frac{1}{m}})$, which is worse than that of UD. Furthermore, as discussed in Section 3, probably UD has a much lower validation risk than CV via exploiting the gradient information of the optimization landscape. Indeed, we show that CV with random search suffers from the curse of dimensionality (See Theorem 7 in Appendix E). Namely, CV requires exponentially large $T$ w.r.t. the dimensionality of the $\lambda$ to achieve a reasonably low empirical risk. The above analysis may explain the superior performance of UD [24], especially when we have a reasonable choice of $T$ and $K$, a sufficiently large $m$ and a sufficiently high-dimensional hyperparameter space.

### 4.3 The Regularized UD Algorithm

Building upon the above theoretical results and analysis, we further investigate how to improve the stability of the UD algorithm via adding regularization. Besides the commonly used weight decay term on the parameter in the inner level, we also employ a similar one on the hyperparameter in the outer level. Formally, the regularized bilevel programming problem is given by:

$$\lambda^*(S^{tr}, S^{val}) = \underbrace{\underset{\lambda \in \Lambda}{\arg\min}\, \hat{R}^{val}(\lambda, \theta^*(\lambda, S^{tr}), S^{val}) + \frac{\mu}{2}||\lambda||_2^2}_{\text{Regularized outer level optimization}},$$

$$\text{where } \theta^*(\lambda, S^{tr}) = \underbrace{\underset{\theta \in \Theta}{\arg\min}\, \hat{R}^{tr}(\lambda, \theta, S^{tr}) + \frac{\nu}{2}||\theta||_2^2}_{\text{Regularized inner level optimization}}, \quad (6)$$

where $\mu$ and $\nu$ are coefficients of the regularization terms. Similar to Algorithm 1, we use UD to approximate Eq. (6). We formally analyze the effect of the regularization terms in both levels (See Theorem 2 and Theorem 3 in Appendix A). In summary, both regularization terms can increase the stability of the UD algorithm, namely, decreasing $\beta$ in a certain way. In particular, the regularization in the outer level decreases $\kappa$ in Theorem 2 while the regularization in the inner level decreases $L$ and $\gamma$ in Theorem 3. Therefore, we can probably obtain a better generalization guarantee by adding regularization in both levels, assuming that the terms do not hurt the validation risk too much.

## 5 Related work

**Hyperparamter optimization.** In addition to UD [9, 10, 11, 29, 35] and CV [2, 31] analyzed in this work, there are alternative approaches that can solve the bilevel programming problem in HO approximately, including implicit gradient [1, 32, 26, 27], Bayesian optimization [38, 21] and hypernetworks [25, 28]. Implicit gradient methods directly estimate the gradient of the outer level problem in Eq. (3), which generally involves an iterative procedure such as conjugate gradient [32] and Neumann approximation [26] to estimate the inverse of a Hessian matrix. [27] also shows that the identity matrix can be a good approximation of the Hessian matrix. Bayesian optimization [38, 21] views the outer level problem of Eq. (4) as a black-box function sampled from a Gaussian process (GP) and updates the posterior of the GP as Eq. (4) is evaluated for new hyperparameters. Hypernetworks [25, 28] learn a proxy function that outputs an approximately optimal hypothesis given a hyperparameter. Besides, UD [9, 10, 11, 29, 35] also has variants which are more time and memory efficient [35, 11]. [35] proposes a faster version of UD by truncating the differentiation w.r.t. the hyperparameter, while [11] proposes a memory-saving version of UD by approximating the trace of inner level optimization in a linear interpolation scheme.

To our knowledge, the analysis of HO in previous work mainly focuses on convergence [10, 35] and unrolling bias [39] from the optimization perspective. The generalization analysis is largely open, which can be dealt with our stability framework in principle. In fact, in addition to UD [9, 10] analyzed in this work, Theorem 1 is also a potential tool for HO algorithms discussed above [1, 32, 26, 27, 38, 21, 25, 28, 11, 29, 35], upon which future work can be built.

**Bilevel Programming.** The bilevel programming is extensively studied from the perspective of optimization. [12, 13, 20] analyze the convergence rate of different approximation methods (e.g., UD

and implicit gradient) when the inner level problem is strongly convex. [19] analyze the complexity lower bounds of bilevel programming when the lower level problem is strongly convex. [18] analyze the convergence rate of bilevel programming under the setting of meta-learning.

**Stability.** The traditional stability theory [3] builds generalization bound of a learning algorithm by studying its stability w.r.t. changing one data point in the training set. It has various extensions. [7, 15] extend the stability for randomized algorithms. [7] focuses on the random perturbations of the cost function. [15] focuses on the randomness in SGD and provides generalization bounds in expectation. Our bound for UD solved by SGD is also in expectation. In such cases, as claimed by [15], *"high probability bounds are hard to obtain by the fact that SGD is itself randomized, and thus a concentration inequality must be devised to account for both the randomness in the data and in the training algorithm"*. The stability notion is also extended to meta-learning [30, 4], where the stability is analyzed w.r.t. changing one meta-dataset to bound the transfer risk on unseen tasks. This work differs from previous extensions in that it considers the stability w.r.t. changing one data point in the validation set, which leads to a generalization bound w.r.t. empirical risk on the validation set.

## 6 Experiments

We conduct experiments to validate our theoretical findings, which are three-folded as follows:

1. We reproduce the mysterious behaviours of UD (e.g., overfitting to the validation data) observed in [10] and attempt to explain them via Theorem 2 and Theorem 3.

2. We empirically compare UD and CV and analyze their performance from the perspective of the expectation bounds in Theorem 2 and Theorem 4.

3. We show the promise of the regularization terms in both levels to validate Theorem 2 and Theorem 3 in Appendix A.

### 6.1 Experimental settings

In our experiments, we consider two widely used tasks, *feature learning* [10] and *data reweighting for noisy labels* [35]. Please refer to Section 2 for the formulation of the two tasks.

In feature learning, we evaluate all algorithms on the Omniglot dataset [22] following [10]. Omniglot consists of grayscale images, and we resize them to $28 \times 28$. The images are symbols from different alphabets. We randomly select 100 classes and obtain a training, validation and testing set of size 500, 100, and 1000 respectively. $\lambda$ represents the parameters in a linear layer of size $784 \rightarrow 256$ following the input $x$. $\theta$ represents the parameters in a MLP of size $256 \rightarrow 128 \rightarrow 100$ to predict the label $y$. We employ a mini-batch version of SGD in both levels of UD with a learning rate $0.1$ and batch size $50$.

In data reweighting, we evaluate all algorithms on the MNIST dataset [23] following [35]. MNIST consists of grayscale hand-written digits of size $28 \times 28$. We randomly select 2000, 200, and 1000 images for training, validation and testing respectively. The label of a training sample is replaced by a uniformly sampled wrong label with probability $0.5$. $\lambda$ represents the logits of the weights of the training data, and $\theta$ represents the parameters in an MLP of size $784 \rightarrow 256 \rightarrow 10$. We employ a mini-batch version of SGD in both levels of UD with a batch size $100$. The learning rate is $10$ in the outer level and $0.3$ in the inner level.

By default, CV shares the same inner loop as UD for a fair comparison in both settings. We tune learning rates and coefficients of the weight decay on another randomly selected set of the same size as the validation set. To eliminate randomness, we average over 5 different runs and report the mean results with the standard deviations in all experiments. Each experiment takes at most 10 hours in one GeForce GTX 1080 Ti GPU.

### 6.2 Trade-off on the values of $T$ and $K$ in UD

Figure 1 presents the results of UD in the feature learning (FL) and data reweighting (DR) tasks, with different values of $K$ and $T$. On the one hand, in both tasks, UD with a large $K$ (e.g., 256 in FL) and a large $T$ overfits the validation data. Namely, the validation loss continues decreasing while the testing loss increases. On the other hand, a small value of $K$ (e.g., 1 in FL) and $T$ will result in

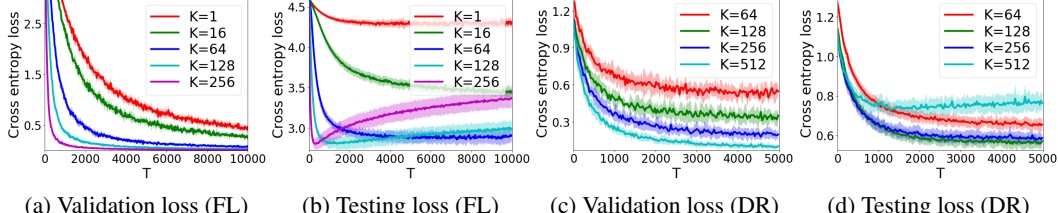

(a) Validation loss (FL)    (b) Testing loss (FL)    (c) Validation loss (DR)    (d) Testing loss (DR)

Figure 1: Results of UD in feature learning (FL) and data reweighting (DR). In both settings, the performance of UD is sensitive to the values of $K$ and $T$. We plot the generalization gap in Appendix G.1.

underfitting to the validation data. The trade-off on the values of $T$ and $K$ agree with our analysis in Theorem 2 and Theorem 3.

We also try a smaller learning rate in the inner level and get a similar overfitting phenomenon of UD (see results in Appendix G.3). In such a case, we use a larger $K$. For instance, a learning rate of $\eta = 0.1$ requires $K = 1024$ inner iterations to overfit. This can be explained by our Theorem 3, which implies that a smaller $\eta$ requires a larger $K$ to make the generalization gap unchanged.

## 6.3    Comparison between CV and UD

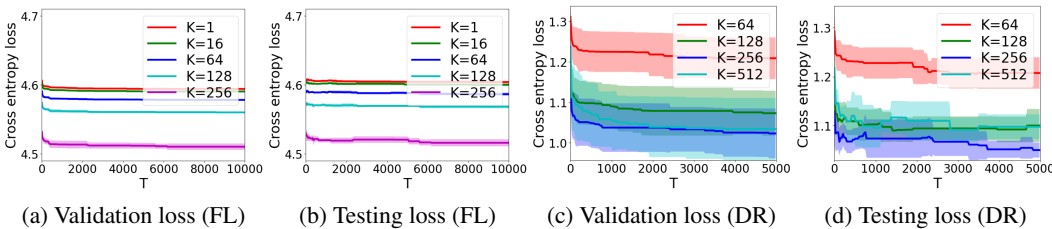

(a) Validation loss (FL)    (b) Testing loss (FL)    (c) Validation loss (DR)    (d) Testing loss (DR)

Figure 2: Results of CV in feature learning (FL) and data reweighting (DR). We do not observe the overfitting phenomenon of CV in all settings. We plot the generalization gap in Appendix G.1.

Figure 2 presents the results of CV in the feature learning (FL) and data reweighting (DR) tasks. First, unlike UD, we do not observe the overfitting phenomenon when using a large $T$ and $K$. This corroborates our analysis in Theorem 4, which claims that the generalization gap of CV grows logarithmically w.r.t. $T$ and does not explicitly depends on $K$. Second, we note that the validation loss of CV is clearly higher than that of UD in Figure 1, which may explain the relatively worse testing loss of CV. Lastly, in FL, the validation loss of CV does not decrease clearly using up to 10,000 hyperparameters. This is because the dimensionality of the hyperparameter is around 200,000, which is too large for CV to optimize, as suggested in Theorem 7 in Appendix E.

We also compare between CV and UD with a smaller number of hyperparameters (see results in Appendix G.4). In this case, the validation losses of UD and CV are comparable and both algorithms fit well on the validation data. However, UD overfits much severely, leading to a worse testing loss than CV. The results agree with our theory.

## 6.4    Effects of Regularization in Both Levels of UD

Figure 3 presents the results of the regularized UD algorithms where the weight decay term is added in the outer (referred to as *Reg-outer*) or the inner level (referred to as *Reg-inner*). We observe that in most of the cases, both Reg-outer and Reg-inner can individually relieve the overfitting problems of UD. Further, within a range, the larger the coefficients of the weight decay terms, the better the results. Such behaviours confirm our Theorem 2 and Theorem 3 in Appendix A. We note that if the regularization is too heavy, for instance, $\nu = 0.1$ in Panel (b) Figure 3, the optimization might be unstable. This suggests another trade-off on determining the values of $\mu$ and $\nu$.

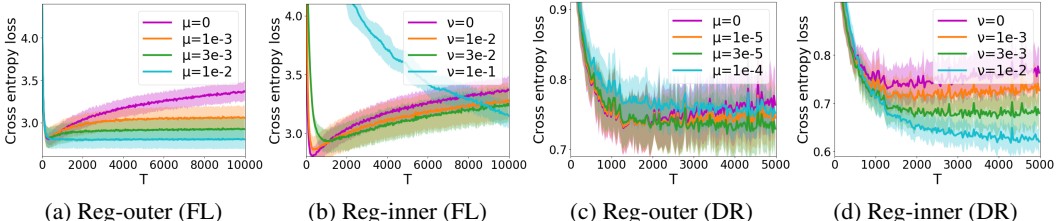

(a) Reg-outer (FL)    (b) Reg-inner (FL)    (c) Reg-outer (DR)    (d) Reg-inner (DR)

Figure 3: Testing loss of the regularized UD algorithms in feature learning (FL) and data reweighting (DR). *Reg-outer* and *Reg-inner* refer to adding regularization individually in the outer and inner levels of UD respectively. We set $K = 256$ in FL and $K = 512$ in DR. In most of the cases, both *Reg-outer* and *Reg-inner* can relieve overfitting in UD but overall there is no clear winner.

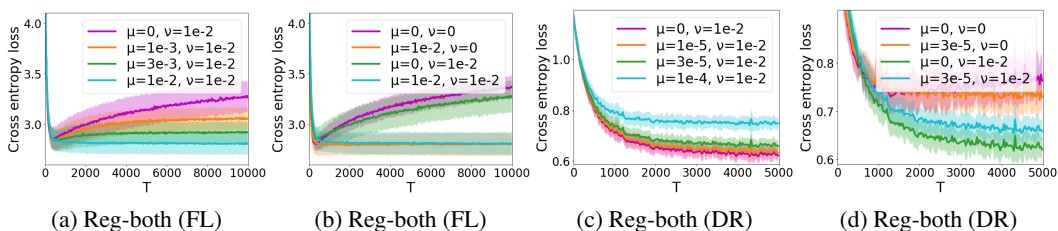

(a) Reg-both (FL)    (b) Reg-both (FL)    (c) Reg-both (DR)    (d) Reg-both (DR)

Figure 4: Testing loss of the regularized UD algorithms in feature learning (FL) and data reweighting (DR). *Reg-both* refers to adding regularization in both the outer and inner levels of UD. We set $K = 256$ in FL and $K = 512$ in DR. In most of the cases, Reg-both is slightly worse than the winner of Reg-outer and Reg-inner.

According to Figure 3, considering the two tasks together, there is no clear winner of Reg-inner and Reg-outer overall. In fact, our Theorem 2 and Theorem 3 in Appendix A show that they influence $\beta$ in incomparable ways. We further apply the two weight decay terms at the same time (referred to as *Reg-both*), and the results are demonstrated in Figure 4. We find that Reg-both is slightly worse than the winner of Reg-outer and Reg-inner. We hypothesize that if one of the weight decay terms can successfully relieve the overfitting problem, then adding another may hurt the final generalization performance because of a higher validation loss. A deeper analysis of the issue is left as future work.

## 7   Conclusion

The paper attempts to understand the generalization behaviour of approximate algorithms to solve the bilevel programming problem in hyperparameter optimization. In particular, we establish an expectation bound for the unrolled differentiation algorithm based on a notion of uniform stability on validation. Our results can explain some mysterious behaviours of the bilevel programming in practice, for instance, overfitting to the validation set. We also present an expectation bound of the classical cross-validation algorithm. Our results suggest that unrolled differentiation algorithms can be better than cross-validation in a theoretical perspective under certain conditions. Furthermore, we prove that regularization terms in both the outer and inner levels can relieve the overfitting problem in the unrolled differentiation algorithm. In experiments on feature learning and data reweighting for noisy labels, we corroborate our theoretical findings.

As an early theoretical work in this area, we find some interesting problems unsolved in this paper, which may inspire future work. First, we do not consider the implicit gradient algorithm, which is an alternative approach to unrolled differentiation and can be analyzed in the stability framework in principle. Second, the comparison between the weight decay terms in different levels is not clear yet. Third, in Theorem 2, we assume the learning rate in the outer level is $\mathcal{O}(\frac{1}{t})$ as in [15], which is a gap between our analysis and the practice.

## Acknowledgements

We thank Yuhao Zhou for valuable feedback on our work. This work was supported by NSFC Projects (Nos. 61620106010, 62061136001, 61621136008, U1811461, U19B2034, U19A2081), Beijing NSF Project (No. JQ19016), Tsinghua-Bosch Joint Center for Machine Learning, Beijing Academy of Artificial Intelligence (BAAI), a grant from Tsinghua Institute for Guo Qiang, Tiangong Institute for Intelligent Computing, and the NVIDIA NVAIL Program with GPU/DGX Acceleration.

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
