# Stability and Generalization of Bilevel Programming in Hyperparameter Optimization: Appendix

**Fan Bao**[*], **Guoqiang Wu**[*†], **Chongxuan Li**[*†], **Jun Zhu**[‡], **Bo Zhang**

Dept. of Comp. Sci. & Tech., Institute for AI, Tsinghua-Huawei Joint Center for AI
BNRist Center, State Key Lab for Intell. Tech. & Sys., Tsinghua University, Beijing, China
bf19@mails.tsinghua.edu.cn,{guoqiangwu90, chongxuanli1991}@gmail.com,
{dcszj, dcszb}@tsinghua.edu.cn

## A    Proofs of Main Theoretical Results

### A.1    Proof of Theorem 1

**Theorem 1** (Generalization bound of a uniformly stable algorithm). *Suppose a randomized HO algorithm* $\mathbf{A}$ *is* $\beta$-*uniformly stable on validation in expectation, then*

$$|\mathbb{E}_{\mathbf{A},S^{tr}\sim(D^{tr})^n,S^{val}\sim(D^{val})^m}\left[R(\mathbf{A}(S^{tr},S^{val}),D^{val})-\hat{R}^{val}(\mathbf{A}(S^{tr},S^{val}),S^{val})\right]|\leq\beta.$$

*Proof.*

$$\begin{aligned}
&|\mathbf{E}_{\mathbf{A},S^{tr},S^{val}}[R(\mathbf{A}(S^{tr},S^{val}),D^{val})-\hat{R}^{val}(\mathbf{A}(S^{tr},S^{val}),S^{val})]|\\
=&|\mathbf{E}_{\mathbf{A},S^{tr},S^{val},z\sim D^{val}}\left[\ell(\mathbf{A}(S^{tr},S^{val}),z)-\ell(\mathbf{A}(S^{tr},S^{val}),z_1^{val})\right]|\\
=&|\mathbf{E}_{\mathbf{A},S^{tr},S^{val},z\sim D^{val}}\left[\ell(\mathbf{A}(S^{tr},z,z_2^{val},\cdots,z_m^{val}),z_1^{val})-\ell(\mathbf{A}(S^{tr},S^{val}),z_1^{val})\right]|\\
\leq&\mathbf{E}_{S^{tr},S^{val},z\sim D^{val}}|\mathbf{E}_{\mathbf{A}}\left[\ell(\mathbf{A}(S^{tr},z,z_2^{val},\cdots,z_m^{val}),z_1^{val})-\ell(\mathbf{A}(S^{tr},S^{val}),z_1^{val})\right]|\leq\beta,
\end{aligned}$$

where the last inequality is due to the definition of stability. ☐

### A.2    Proof of Theorem 2

Here we prove a more general version of Theorem 2 in the full paper by considering SGD with weight decay in the outer level, i.e.,

$$\lambda_{t+1}=(1-\alpha_{t+1}\mu)\lambda_t-\alpha_{t+1}\nabla_{\lambda_t}\ell(\lambda_t,\hat{\theta}(\lambda_t,S^{tr}),z_j^{val}), \tag{1}$$

where $\alpha_t$ is the learning rate, $\mu$ is the weight decay, $j$ is randomly selected from $\{1,\cdots,m\}$ and $\hat{\theta}$ is a random function. Theorem 2 in the full paper can be simply derived by letting $\mu=0$.

**Theorem 2** (Uniform stability of algorithms with SGD in the outer level). *Suppose* $\hat{\theta}$ *is a random function in a function space* $\mathcal{G}_{\hat{\theta}}$ *and* $\forall S^{tr}\in Z^n$, $\forall z\in Z$, $\forall g\in\mathcal{G}_{\hat{\theta}}$, $\ell(\lambda,g(\lambda,S^{tr}),z)$ *as a function of* $\lambda$ *is* $L$-*Lipschitz continuous and* $\gamma$-*Lipschitz smooth, let* $c\leq\frac{s(\ell)}{2L^2}$, $\mu\leq\min(\frac{1}{c},(1-1/m)\gamma)$ *and* $\kappa=\frac{c((1-1/m)\gamma-\mu)}{c((1-1/m)\gamma-\mu)+1}$. *Then, solving Eq. (4) in the full paper with* $T$ *steps SGD, learning rate* $\alpha_t\leq\frac{c}{t}$ *and weight decay* $\mu$ *in the outer level is* $\beta$-*uniformly stable on validation in expectation with*

$$\beta=\frac{2cL^2}{m}\left(\frac{1}{\kappa}\left(\left(\frac{Ts(\ell)}{2cL^2}\right)^{\kappa}-1\right)+1\right),$$

---

[*]Equal contribution

[†]G. Wu is now at School of Software, Shandong University and C. Li is now at Gaoling School of AI, Renmin University of China. The work was done when they were at Tsinghua University.

[‡]Corresponding author.

35th Conference on Neural Information Processing Systems (NeurIPS 2021).

*which is increasing w.r.t. $L$, $\gamma$ and decreasing w.r.t. $\mu$.*

*Proof.* Suppose $S^{tr} \in Z^n$ and $z \in Z$, let $f(\lambda, g) = \ell(\lambda, g(\lambda, S^{tr}), z)$, where we omit the dependency on $S^{tr}$ and $z$ for simplicity, then $f(\lambda, g)$ is as a function of $\lambda$ is $L$-Lipschitz continuous and $\gamma$-Lipschitz smooth. Suppose $S^{val}$ and $S'^{val}$ differ in at most one point, let $\{\lambda_t\}_{t \geq 0}$ and $\{\lambda'_t\}_{t \geq 0}$ be the trace of Eq. (1) with $S^{val}$ and $S'^{val}$ respectively. Then the output of the HO algorithm $\mathbf{A}$ with $t$ steps SGD in the outer level is

$$\mathbf{A}(S^{tr}, S^{val}) = (\lambda_t, \hat{\theta}(\lambda_t, S^{tr})), \ \mathbf{A}(S^{tr}, S'^{val}) = (\lambda'_t, \hat{\theta}(\lambda'_t, S^{tr})),$$

and

$$\ell(\mathbf{A}(S^{tr}, S^{val}), z) = \ell(\lambda_t, \hat{\theta}(\lambda_t, S^{tr}), z) = f(\lambda_t, \hat{\theta}),$$
$$\ell(\mathbf{A}(S^{tr}, S'^{val}), z) = \ell(\lambda'_t, \hat{\theta}(\lambda'_t, S^{tr}), z) = f(\lambda'_t, \hat{\theta}).$$

Let $\delta_t = ||\lambda_t - \lambda'_t||$. Suppose $0 \leq t_0 \leq t$, we have

$$\mathbf{E}\left[|f(\lambda_t, \hat{\theta}) - f(\lambda'_t, \hat{\theta})|\right] = \mathbf{E}\left[|f(\lambda_t, \hat{\theta}) - f(\lambda'_t, \hat{\theta})| \cdot 1_{\delta_{t_0}=0}\right]$$
$$+ \mathbf{E}\left[|f(\lambda_t, \hat{\theta}) - f(\lambda'_t, \hat{\theta})| \cdot 1_{\delta_{t_0}>0}\right]$$
$$\leq L\mathbf{E}\left[\delta_t \cdot 1_{\delta_{t_0}=0}\right] + P(\delta_{t_0} > 0)s(\ell).$$

Without loss of generality, we assume $S^{val}$ and $S'^{val}$ at most differ in at the first point. If SGD doesn't selects the first point for the first $t_0$ iterations, then $\delta_{t_0} = 0$. As a result,

$$P(\delta_{t_0} = 0) \geq (1 - \frac{1}{m})^{t_0} \geq 1 - \frac{t_0}{m}.$$

Therefore, $P(\delta_{t_0} > 0) \leq \frac{t_0}{m}$ and we have

$$\mathbf{E}\left[|f(\lambda_t, \hat{\theta}) - f(\lambda'_t, \hat{\theta})|\right] \leq L\mathbf{E}\left[\delta_t \cdot 1_{\delta_{t_0}=0}\right] + \frac{t_0}{m}s(\ell). \tag{2}$$

Now we bound $\mathbf{E}\left[\delta_t \cdot 1_{\delta_{t_0}=0}\right]$. Let $\gamma' = (1 - 1/m)\gamma - \mu$ and let $j$ be the index selected by SGD at the $t + 1$ iteration, then we have

$$\mathbf{E}\left[\delta_{t+1} \cdot 1_{\delta_{t_0}=0}\right] \leq \mathbf{E}\left[\delta_{t+1} \cdot 1_{j=1} \cdot 1_{\delta_{t_0}=0}\right] + \mathbf{E}\left[\delta_{t+1} \cdot 1_{j>1} \cdot 1_{\delta_{t_0}=0}\right]$$
$$\leq \frac{1}{m}(|1 - \alpha_{t+1}\mu| \cdot \mathbf{E}[\delta_t \cdot 1_{\delta_{t_0}=0}] + 2\alpha_{t+1}L)$$
$$+ \frac{m-1}{m}(|1 - \alpha_{t+1}\mu| + \alpha_{t+1}\gamma)\mathbf{E}[\delta_t \cdot 1_{\delta_{t_0}=0}]$$
$$= (1 + \alpha_{t+1}\gamma')\mathbf{E}[\delta_t \cdot 1_{\delta_{t_0}=0}] + \frac{2\alpha_{t+1}L}{m}$$
$$\leq \exp(\alpha_{t+1}\gamma')\mathbf{E}[\delta_t \cdot 1_{\delta_{t_0}=0}] + \frac{2\alpha_{t+1}L}{m}$$
$$\leq \exp(\frac{c}{t+1}\gamma')\mathbf{E}[\delta_t \cdot 1_{\delta_{t_0}=0}] + \frac{2cL}{(t+1)m}.$$

As a result,

$$\mathbf{E}[\delta_t \cdot 1_{\delta_{t_0}=0}] \leq \sum_{j=t_0+1}^{t} \frac{2cL}{jm} \prod_{k=j+1}^{t} \exp(\frac{c\gamma'}{k}) = \sum_{j=t_0+1}^{t} \frac{2cL}{jm} \exp(c\gamma' \sum_{k=j+1}^{t} \frac{1}{k})$$
$$\leq \sum_{j=t_0+1}^{t} \frac{2cL}{jm} \exp(c\gamma' \ln\frac{t}{j}) = \sum_{j=t_0+1}^{t} \frac{2cL}{jm} \left(\frac{t}{j}\right)^{c\gamma'}$$
$$= \frac{2cLt^{c\gamma'}}{m} \sum_{j=t_0+1}^{t} \left(\frac{1}{j}\right)^{1+c\gamma'} \leq \frac{2cLt^{c\gamma'}}{m} \frac{t^{-c\gamma'} - t_0^{-c\gamma'}}{-c\gamma'}$$
$$= \frac{2L}{m\gamma'} \left(\left(\frac{t}{t_0}\right)^{c\gamma'} - 1\right).$$

Combining with Eq. (2), we have

$$\mathbf{E}\left[|f(\lambda_T, \hat{\theta}) - f(\lambda'_T, \hat{\theta})|\right] \leq \inf_{0 \leq t_0 \leq T} \frac{2L^2}{m\gamma'} \left(\left(\frac{T}{t_0}\right)^{c\gamma'} - 1\right) + \frac{t_0}{m} s(\ell). \tag{3}$$

The right hand side is approximately minimized when

$$t_0 = \left(\frac{2cL^2}{s(\ell)}\right)^{\frac{1}{c\gamma'+1}} T^{\frac{c\gamma'}{c\gamma'+1}} \leq T,$$

which gives

$$\mathbf{E}\left[|f(\lambda_T, \hat{\theta}) - f(\lambda'_T, \hat{\theta})|\right] \leq \frac{1 + 1/c\gamma'}{m} (2cL^2)^{\frac{1}{c\gamma'+1}} T^{\frac{c\gamma'}{c\gamma'+1}} (s(\ell))^{\frac{c\gamma'}{c\gamma'+1}} - \frac{2L^2}{m\gamma'} =: \beta.$$

Let $\kappa = \frac{c\gamma'}{c\gamma'+1} = \frac{c((1-1/m)\gamma-\mu)}{c((1-1/m)\gamma-\mu)+1}$, then $\beta$ can be written as

$$\beta = \frac{2cL^2}{m}\left(\frac{1}{\kappa}\left(\left(\frac{Ts(\ell)}{2cL^2}\right)^{\kappa} - 1\right) + 1\right).$$

Since the r.h.s. of Eq. (3) is increasing w.r.t. $L$ and $\gamma'$, where $\gamma'$ is further increasing w.r.t. $\gamma$ and decreasing w.r.t. $\mu$, we can conclude $\beta$ is increasing w.r.t. $L, \gamma$ and decreasing w.r.t. $\mu$. □

### A.3 Proof of Theorem 3

**Definition 1.** *(Lipschitz continuous) Suppose $(X, d_X), (Y, d_Y)$ are two metric spaces and $f : X \to Y$. We define $f$ is $L$ Lipschitz continuous iff $\forall a, b \in X, d_Y(f(a), f(b)) \leq Ld_X(a, b)$.*

**Definition 2.** *(Lipschitz smooth) Suppose $X, Y$ are subsets of two real normed vector spaces and $f : X \to Y$ is differentiable. We define $f$ is $\gamma$ Lipschitz smooth iff $f'$ is $\gamma$ Lipschitz continuous.*

**Definition 3.** *(Lipschitz norm) Suppose $(X, d_X), (Y, d_Y)$ are two metric spaces, $f : X \to Y$, we define $||f||_{Lip} \triangleq \inf\{L \in [0, \infty] : \forall a, b \in X, d_Y(f(a), f(b)) \leq Ld_X(a, b)\}$, i.e., the minimum $L$ such that $f$ is $L$ Lipschitz continuous.*

**Definition 4.** *Given a function $f(\lambda, \theta)$, we use $||f(\lambda, \theta)||_{\lambda \in \Lambda, Lip}$ and $||f(\lambda, \theta)||_{\theta \in \Theta, Lip}$ to explicitly denote the Lipschitz norm of $f$ w.r.t. $\lambda \in \Lambda$ and $\theta \in \Theta$ respectively.*

**Definition 5.** *(Vector norm) Suppose $a \in \mathbf{R}^m$, we use $||a||$ to denote the $l_2$ norm of $a$.*

**Definition 6.** *(Matrix norm) Suppose $A \in \mathbf{R}^{m \times n}$, we define $||A|| \triangleq \sup_{0 \neq a \in \mathbf{R}^m} \frac{||Aa||}{||a||}$, i.e., the norm of the linear operator induced by $A$.*

**Lemma 1.** *Suppose $X, Y$ are two real normed vector spaces, $\Omega$ is an open set of $X$, $f : \Omega \to Y$ is continuously differentiable, $S \subset \Omega$ is convex and has non-empty interior, then $||f|_S||_{Lip} = \sup_{c \in S} ||f'(c)||$.*

*Proof.* Suppose $a, b \in S$, according to the mean value theorem, there is a $c$ lies in the segment determined by $a$ and $b$, s.t., $||f(b) - f(a)|| \leq ||f'(c)(b-a)||$. Furthermore, we have

$$||f'(c)(b-a)|| \leq ||f'(c)|| \cdot ||b-a|| \leq \sup_{c \in S} ||f'(c)|| \cdot ||b-a||.$$

Thereby, $f|_S$ is $\sup_{c \in S} ||f'(c)||$ Lipschitz continuous and $||f|_S||_{Lip} \leq \sup_{c \in S} ||f'(c)||$.

Suppose $c \in S^\circ$, where $S^\circ$ is the interior of $S$ and $u \in X$ with $||u|| = 1$, then

$$\lim_{\epsilon \to 0} \frac{f(c + \epsilon u) - f(c)}{\epsilon} = f'(c)u.$$

Thereby,

$$||f|_S||_{Lip} \geq \lim_{\epsilon \to 0} ||\frac{f(c + \epsilon u) - f(c)}{\epsilon}|| = ||f'(c)u||.$$

Since $u$ is arbitrary, we have $||f'(c)|| = \sup\limits_{u \in X, ||u||=1} ||f'(c)u|| \leq ||f|_S||_{Lip}$.

Since $S$ has non-empty interior, we have $S \subset \overline{S^\circ}$ by the property of convex sets. Suppose $c \in S$, then $c \in \overline{S^\circ}$ and there is a sequence $c_n \in S^\circ$, s.t., $c_n \to c$. Since $c_n \in S^\circ$, we have $||f'(c_n)|| \leq ||f|_S||_{Lip}$. Let $n \to \infty$, by the continuity of $f'$, we have $||f'(c)|| \leq ||f|_S||_{Lip}$. Since $c \in S$ is arbitrary, we have $\sup\limits_{c \in S} ||f'(c)|| \leq ||f|_S||_{Lip}$. Finally, we have $\sup\limits_{c \in S} ||f'(c)|| = ||f|_S||_{Lip}$. $\qquad\square$

**Lemma 2.** *Suppose $\Lambda$ and $\Theta$ are convex and compact with non-empty interiors, $Z$ is compact, $\Lambda \times \Theta \times Z$ is included in an open set $\Omega$ and $f(\lambda, \theta, z) \in C^k(\Omega)$, then for all $i \leq k - 1$ order partial differential $h(\lambda, \theta, z)$ of $f(\lambda, \theta, z)$, we have $\sup\limits_{\theta \in \Theta, z \in Z} ||h(\lambda, \theta, z)||_{\lambda \in \Lambda, Lip} < \infty$ and $\sup\limits_{\lambda \in \Lambda, z \in Z} ||h(\lambda, \theta, z)||_{\theta \in \Theta, Lip} < \infty$.*

*Proof.* Suppose $h(\lambda, \theta, z)$ is a $i \leq k - 1$ order partial differential of $f(\lambda, \theta, z)$, then $h(\lambda, \theta, z) \in C^1(\Omega)$ and $\nabla_\lambda h(\lambda, \theta, z) \in C(\Omega)$. Since $\Lambda \times \Theta \times Z$ is compact, $\nabla_\lambda h(\lambda, \theta, z)$ is bounded in $\Lambda \times \Theta \times Z$. According to Lemma 1, we have

$$\sup\limits_{\theta \in \Theta, z \in Z} ||h(\lambda, \theta, z)||_{\lambda \in \Lambda, Lip} = \sup\limits_{\theta \in \Theta, z \in Z} \sup\limits_{\lambda \in \Lambda} ||\nabla_\lambda h(\lambda, \theta, z)|| < \infty.$$

Similarly, we can derive $\sup\limits_{\lambda \in \Lambda, z \in Z} ||h(\lambda, \theta, z)||_{\theta \in \Theta, Lip} < \infty$. $\qquad\square$

**Lemma 3.** *Suppose (1) $\forall 1 \leq k \leq K$, $\forall \lambda \in \Lambda$, $G_{\lambda,k}(\theta)$ is a mapping from $\Theta$ to $\Theta$, i.e., $G_{\lambda,k} : \Theta \to \Theta$, (2) $\forall 1 \leq k \leq K$, $\forall \theta \in \Theta$, $G_{\lambda,k}(\theta)$ as a function of $\lambda$ is $L_1^G < \infty$ Lipschitz continuous, (3) $\forall 1 \leq k \leq K$, $\forall \lambda \in \Lambda$, $G_{\lambda,k}(\theta)$ as a function of $\theta$ is $L_2^G < \infty$ Lipschitz continuous. Let $\hat{\theta}(\lambda) = G_{\lambda,K}(G_{\lambda,K-1}(\cdots(G_{\lambda,1}(\theta_0))))$, then $\hat{\theta}(\lambda)$ is $L^{\hat{\theta}}$ Lipschitz continuous with*

$$L^{\hat{\theta}} = \begin{cases} L_1^G \frac{(L_2^G)^K - 1}{L_2^G - 1} & L_2^G \neq 1 \\ K L_1^G & L_2^G = 1 \end{cases}.$$

*Proof.* We use $\theta_K(\lambda)$ to denote $G_{\lambda,K}(G_{\lambda,K-1}(\cdots(G_{\lambda,1}(\theta_0))))$. Suppose $\lambda, \lambda' \in \Lambda$ and $K \geq 1$, we have

$$||\theta_K(\lambda) - \theta_K(\lambda')|| = ||G_{\lambda,K}(\theta_{K-1}(\lambda)) - G_{\lambda',K}(\theta_{K-1}(\lambda'))||$$
$$\leq ||G_{\lambda,K}(\theta_{K-1}(\lambda)) - G_{\lambda',K}(\theta_{K-1}(\lambda))|| + ||G_{\lambda',K}(\theta_{K-1}(\lambda)) - G_{\lambda',K}(\theta_{K-1}(\lambda'))||$$
$$\leq L_1^G ||\lambda - \lambda'|| + L_2^G ||\theta_{K-1}(\lambda) - \theta_{K-1}(\lambda')||.$$

If $L_2^G \neq 1$, we have $||\theta_K(\lambda) - \theta_K(\lambda')|| \leq \frac{(L_2^G)^K - 1}{L_2^G - 1} L_1^G ||\lambda - \lambda'||$.

If $L_2^G = 1$, we have $||\theta_K(\lambda) - \theta_K(\lambda')|| \leq K L_1^G ||\lambda - \lambda'||$. $\qquad\square$

**Lemma 4.** *Suppose (1) $\forall 1 \leq k \leq K$, $\forall \lambda \in \Lambda$, $G_{\lambda,k}(\theta)$ is a mapping from $\Theta$ to $\Theta$, i.e., $G_{\lambda,k} : \Theta \to \Theta$, (2) $\forall 1 \leq k \leq K$, $\forall \theta \in \Theta$, $G_{\lambda,k}(\theta)$ and $\frac{\partial}{\partial \lambda} G_{\lambda,k}(\theta)$ as a function of $\lambda$ is $L_1^G$ and $\gamma_1^G$ Lipschitz continuous respectively, (3) $\forall 1 \leq k \leq K$, $\forall \lambda \in \Lambda$, $G_{\lambda,k}(\theta)$ and $\frac{\partial}{\partial \theta} G_{\lambda,k}(\theta)$ as a function of $\theta$ is $L_2^G$ and $\gamma_2^G$ Lipschitz continuous respectively, (4) $\forall 1 \leq k \leq K$, $\forall \theta \in \Theta$, $\frac{\partial}{\partial \theta} G_{\lambda,k}(\theta)$ as a function of $\lambda$ is $\gamma_3^G \geq 0$ Lipschitz continuous, (5) $\forall 1 \leq k \leq K$, $\forall \lambda \in \Lambda$, $\frac{\partial}{\partial \lambda} G_{\lambda,k}(\theta)$ as a function of $\theta$ is $\gamma_4^G \geq 0$ Lipschitz continuous. Let $\hat{\theta}(\lambda) = G_{\lambda,K}(G_{\lambda,K-1}(\cdots(G_{\lambda,1}(\theta_0))))$, then $\hat{\theta}(\lambda)$ is $\gamma^{\hat{\theta}}$ Lipschitz smooth with*

$$\gamma^{\hat{\theta}} = \begin{cases} \mathcal{O}((L_2^G)^{2K}) & L_2^G > 1 \\ \mathcal{O}(K^3) & L_2^G = 1, L_1^G > 0 \\ \mathcal{O}(K) & L_2^G = 1, L_1^G = 0 \\ \mathcal{O}(1) & L_2^G < 1 \end{cases},$$

*and $\gamma^{\hat{\theta}}$ is determined by $L_1^G, L_2^G, \gamma_1^G, \gamma_2^G, \gamma_3^G, \gamma_4^G, K$.*

*Proof.* Suppose $1 \le k \le K$, we use $\theta_k(\lambda)$ to denote $G_{\lambda,k}(G_{\lambda,k-1}(\cdots (G_{\lambda,1}(\theta_0))))$. According to Lemma 3, $\theta_k(\lambda)$ is $L^{\hat{\theta},k} = \begin{cases} L_1^G \frac{(L_2^G)^k - 1}{L_2^G - 1} & L_2^G \ne 1 \\ kL_1^G & L_2^G = 1 \end{cases}$ Lipschitz continuous. Taking gradient to $\theta_k(\lambda)$ w.r.t. $\lambda$, we have

$$\frac{\partial}{\partial \lambda} \theta_k(\lambda) = \frac{\partial}{\partial \lambda} G_{\lambda,k}(\theta_{k-1}(\lambda)) = \left[ \frac{\partial}{\partial \lambda} G_{\lambda,k}(\theta) \right]\Big|_{\theta = \theta_{k-1}(\lambda)} + \left[ \frac{\partial}{\partial \theta} G_{\lambda,k}(\theta) \right]\Big|_{\theta = \theta_{k-1}(\lambda)} \left[ \frac{\partial}{\partial \lambda} \theta_{k-1}(\lambda) \right].$$

Taking the Lipschitz constant w.r.t. $\lambda$, we have

$$\left|\left| \left[ \frac{\partial}{\partial \lambda} G_{\lambda,k}(\theta) \right]\Big|_{\theta = \theta_{k-1}(\lambda)} \right|\right|_{\lambda,Lip} \le \gamma_1^G + \gamma_4^G L^{\hat{\theta},k-1},$$

$$\left|\left| \left[ \frac{\partial}{\partial \theta} G_{\lambda,k}(\theta) \right]\Big|_{\theta = \theta_{k-1}(\lambda)} \right|\right|_{\lambda,Lip} \le \gamma_3^G + \gamma_2^G L^{\hat{\theta},k-1},$$

$$\begin{aligned}
\left|\left| \frac{\partial}{\partial \lambda} \theta_k(\lambda) \right|\right|_{\lambda,Lip} \le & \left|\left| \left[ \frac{\partial}{\partial \lambda} G_{\lambda,k}(\theta) \right]\Big|_{\theta = \theta_{k-1}(\lambda)} \right|\right|_{\lambda,Lip} \\
& + \left|\left| \left[ \frac{\partial}{\partial \theta} G_{\lambda,k}(\theta) \right]\Big|_{\theta = \theta_{k-1}(\lambda)} \right|\right|_{\lambda,Lip} \sup_{\lambda \in \Lambda} \left|\left| \frac{\partial}{\partial \lambda} \theta_{k-1}(\lambda) \right|\right| \\
& + \sup_{\lambda \in \Lambda, \theta \in \Theta} \left|\left| \frac{\partial}{\partial \theta} G_{\lambda,k}(\theta) \right|\right| \left|\left| \frac{\partial}{\partial \lambda} \theta_{k-1}(\lambda) \right|\right|_{\lambda,Lip} \\
\le & \gamma_1^G + \gamma_4^G L^{\hat{\theta},k-1} + (\gamma_3^G + \gamma_2^G L^{\hat{\theta},k-1}) L^{\hat{\theta},k-1} + L_2^G \left|\left| \frac{\partial}{\partial \lambda} \theta_{k-1}(\lambda) \right|\right|_{\lambda,Lip} \\
= & \gamma_2^G (L^{\hat{\theta},k-1})^2 + (\gamma_3^G + \gamma_4^G) L^{\hat{\theta},k-1} + \gamma_1^G + L_2^G \left|\left| \frac{\partial}{\partial \lambda} \theta_{k-1}(\lambda) \right|\right|_{\lambda,Lip}.
\end{aligned}$$

As for $\theta_0$, we have

$$\left|\left| \frac{\partial}{\partial \lambda} \theta_0(\lambda) \right|\right|_{\lambda,Lip} = 0.$$

Let $\gamma^{\hat{\theta}}$ be the $K$th term of the sequence defined by

$$a_k = \gamma_2^G (L^{\hat{\theta},k-1})^2 + (\gamma_3^G + \gamma_4^G) L^{\hat{\theta},k-1} + \gamma_1^G + L_2^G a_{k-1}, \quad a_0 = 0,$$

which is determined by $L_1^G, L_2^G, \gamma_1^G, \gamma_2^G, \gamma_3^G, \gamma_4^G$, then $||\frac{\partial}{\partial \lambda} \theta_K(\lambda)||_{\lambda,Lip} \le \gamma^{\hat{\theta}}$ and $\hat{\theta}(\lambda) = \theta_K(\lambda)$ is $\gamma^{\hat{\theta}}$ Lipschitz smooth. Finally, we analyze the order of $\gamma^{\hat{\theta}}$. If $L_2^G > 1$, then $L^{\hat{\theta},K} = \mathcal{O}((L_2^G)^K)$ and $\gamma^{\hat{\theta}} = \mathcal{O}((L_2^G)^{2K})$. If $L_2^G = 1$, then $L^{\hat{\theta},K} = KL_1^G + L^{\theta_0}$ and $\gamma^{\hat{\theta}} = \begin{cases} \mathcal{O}(K) & L_1^G = 0 \\ \mathcal{O}(K^3) & L_1^G > 0 \end{cases}$. If $L_2^G < 1$, then $L^{\hat{\theta},K} = \mathcal{O}(1)$ and $\gamma^{\hat{\theta}} = \mathcal{O}(1)$. $\qquad \square$

**Assumption 1.** $\Lambda$ *and* $\Theta$ *are compact and convex with non-empty interiors, and* $Z$ *is compact.*

**Assumption 2.** $\ell(\lambda, \theta, z) \in C^2(\Omega)$, *where* $\Omega$ *is an open set including* $\Lambda \times \Theta \times Z$ *(i.e.,* $\ell$ *is second order continuously differentiable on* $\Omega$*).*

**Assumption 3.** $\varphi_i(\lambda, \theta, z) \in C^3(\Omega)$, *where* $\Omega$ *is an open set including* $\Lambda \times \Theta \times Z$ *(i.e.,* $\varphi_i$ *is third order continuously differentiable on* $\Omega$*).*

**Assumption 4.** $\varphi_i(\lambda, \theta, z)$ *is* $\gamma_\varphi$*-Lipschitz smooth as a function of* $\theta$ *for all* $1 \le i \le n$, $z \in Z$ *and* $\lambda \in \Lambda$ *(Assumption 3 implies such a constant* $\gamma_\varphi$ *exists).*

Here we prove a more general version of Theorem 3 in the full paper by considering SGD or GD with weight decay $\nu$ in the inner level. Theorem 3 in the full paper can be simply derived by letting $\nu = 0$.

**Theorem 3.** *Suppose Assumption 1,2,3,4 hold and the inner level problem is solved with* $K$ *steps SGD or GD with learning rate* $\eta$ *and weight decay* $\nu$, *then* $\forall S^{tr} \in Z^n$, $\forall z \in Z$, $\forall g \in \mathcal{G}_{\hat{\theta}}$, $\ell(\lambda, g(\lambda, S^{tr}), z)$ *as a function of* $\lambda$ *is* $L = \mathcal{O}((1 + \eta(\gamma_\varphi - \nu))^K)$ *Lipschitz continuous and* $\gamma = \mathcal{O}((1 + \eta(\gamma_\varphi - \nu))^{2K})$ *Lipschitz smooth.*

*Proof.* The $k$th updating step of SGD can be written as

$$G_{\lambda,k}(\theta) = (1 - \eta\nu)\theta - \eta\nabla_\theta\varphi_{j_k}(\lambda, \theta, z_{j_k}^{tr}) = \nabla_\theta\left(\frac{(1-\eta\nu)}{2}||\theta||^2 - \eta\varphi_{j_k}(\lambda, \theta, z_{j_k}^{tr})\right),$$

where $j_k$ is randomly selected from $\{1, 2, \cdots, n\}$. The output of $K$ steps SGD is $\hat{\theta}(\lambda, S^{tr}) = G_{\lambda,K}(G_{\lambda,K-1}(\cdots(G_{\lambda,1}(\theta_0))))$ and $\mathcal{G}_{\hat{\theta}}$ is formed by iterates over $(j_1, j_2, \cdots, j_K) \in \{1, 2, \cdots, n\}^K$.

According to Lemma 2 and Assumption 3, we have

$$L_1^G \triangleq \sup_{k,j_k,S^{tr},\theta}||G_{\lambda,k}(\theta)||_{\lambda\in\Lambda,Lip} = \sup_{i,z,\theta}||\nabla_\theta\left(\frac{(1-\eta\nu)}{2}||\theta||^2 - \eta\varphi_i(\lambda, \theta, z)\right)||_{\lambda\in\Lambda,Lip} < \infty.$$

Similarly, we have

$$\gamma_1^G \triangleq \sup_{k,j_k,S^{tr},\theta}||\frac{\partial}{\partial\lambda}G_{\lambda,k}(\theta)||_{\lambda\in\Lambda,Lip} < \infty, \ \gamma_2^G \triangleq \sup_{k,j_k,S^{tr},\lambda}||\frac{\partial}{\partial\theta}G_{\lambda,k}(\theta)||_{\theta\in\Theta,Lip} < \infty,$$

$$\gamma_3^G \triangleq \sup_{k,j_k,S^{tr},\theta}||\frac{\partial}{\partial\theta}G_{\lambda,k}(\theta)||_{\lambda\in\Lambda,Lip} < \infty, \ \gamma_4^G \triangleq \sup_{k,j_k,S^{tr},\lambda}||\frac{\partial}{\partial\lambda}G_{\lambda,k}(\theta)||_{\theta\in\Theta,Lip} < \infty.$$

According to Assumption 4, we have

$$\sup_{k,j_k,S^{tr},\lambda}||G_{\lambda,k}(\theta)||_{\theta\in\Theta,Lip} \leq 1 - \eta\nu + \eta\gamma_\varphi = 1 + \eta(\gamma_\varphi - \nu) \triangleq L_2^G < \infty.$$

According to Lemma 3 and Lemma 4, $\hat{\theta}(\lambda, S^{tr})$ is $L^{\hat{\theta}} = L_1^G\frac{(L_2^G)^K-1}{L_2^G-1}$ Lipschitz continuous and $\gamma^{\hat{\theta}} = \mathcal{O}((L_2^G)^{2K})$ Lipschitz smooth as a function of $\lambda$. By definition, $L^{\hat{\theta}}$ and $\gamma^{\hat{\theta}}$ are independent of the training dataset $S^{tr}$ and the random indices $(j_1, j_2, \cdots, j_K)$ and thereby the randomness of $\hat{\theta}$.

According to Lemma 2 and Assumption 2, we have

$$L_1^\ell = \sup_{\theta\in\Theta,z\in Z}||\ell(\lambda, \theta, z)||_{\lambda\in\Lambda,Lip} < \infty, \ L_2^\ell = \sup_{\lambda\in\Lambda,z\in Z}||\ell(\lambda, \theta, z)||_{\theta\in\Theta,Lip} < \infty.$$

Similarly, we have

$$\gamma_1^\ell \triangleq \sup_{\theta,z}||\left[\frac{\partial}{\partial\lambda}\ell(\lambda, \theta, z)\right]||_{\lambda\in\Lambda,Lip} < \infty, \ \gamma_2^\ell \triangleq \sup_{\lambda,z}||\left[\frac{\partial}{\partial\theta}\ell(\lambda, \theta, z)\right]||_{\theta\in\Theta,Lip} < \infty,$$

$$\gamma_3^\ell \triangleq \sup_{\theta,z}||\left[\frac{\partial}{\partial\theta}\ell(\lambda, \theta, z)\right]||_{\lambda\in\Lambda,Lip} < \infty, \ \gamma_4^\ell \triangleq \sup_{\lambda,z}||\left[\frac{\partial}{\partial\lambda}\ell(\lambda, \theta, z)\right]||_{\theta\in\Theta,Lip} < \infty.$$

Suppose $z \in Z$, firstly we consider the Lipschitz continuity of $\ell(\lambda, \hat{\theta}(\lambda, S^{tr}), z)$:

$$||\ell(\lambda, \hat{\theta}(\lambda, S^{tr}), z)||_{\lambda\in\Lambda,Lip}$$
$$\leq \sup_{\theta\in\Theta,z\in Z}||\ell(\lambda, \theta, z)||_{\lambda\in\Lambda,Lip} + \sup_{\lambda\in\Lambda,z\in Z}||\ell(\lambda, \theta, z)||_{\theta\in\Theta,Lip}\cdot||\hat{\theta}(\lambda, S^{tr})||_{\lambda\in\Lambda,Lip}$$
$$\leq L_1^\ell + L_2^\ell L^{\hat{\theta}} \triangleq L. \tag{4}$$

Then we consider the Lipschitz continuity of $\frac{\partial}{\partial\lambda}\ell(\lambda, \hat{\theta}(\lambda, S^{tr}), z)$, which can be expanded as

$$\frac{\partial}{\partial\lambda}\ell(\lambda, \hat{\theta}(\lambda, S^{tr}), z) = \left[\frac{\partial}{\partial\lambda}\ell(\lambda, \theta, z)\right]\Big|_{\theta=\hat{\theta}(\lambda,S^{tr})} + \left[\frac{\partial}{\partial\theta}\ell(\lambda, \theta, z)\right]\Big|_{\theta=\hat{\theta}(\lambda,S^{tr})}\left[\frac{\partial}{\partial\lambda}\hat{\theta}(\lambda, S^{tr})\right].$$

Taking the Lipschitz norm w.r.t. $\lambda$, we have

$$||\left[\frac{\partial}{\partial\lambda}\ell(\lambda, \theta, z)\right]\Big|_{\theta=\hat{\theta}(\lambda,S^{tr})}||_{\lambda\in\Lambda,Lip} \leq \gamma_1^\ell + \gamma_4^\ell L^{\hat{\theta}},$$

$$|| \left[ \frac{\partial}{\partial \theta} \ell(\lambda, \theta, z) \right] \Big|_{\theta = \hat{\theta}(\lambda, S^{tr})} ||_{\lambda \in \Lambda, Lip} \leq \gamma_3^\ell + \gamma_2^\ell L^{\hat{\theta}},$$

which yields

$$|| \frac{\partial}{\partial \lambda} \ell(\lambda, \hat{\theta}(\lambda, S^{tr}), z)||_{\lambda \in \Lambda, Lip}$$

$$\leq || \left[ \frac{\partial}{\partial \lambda} \ell(\lambda, \theta, z) \right] \Big|_{\theta = \hat{\theta}(\lambda, S^{tr})} ||_{\lambda \in \Lambda, Lip}$$

$$+ || \left[ \frac{\partial}{\partial \theta} \ell(\lambda, \theta, z) \right] \Big|_{\theta = \hat{\theta}(\lambda, S^{tr})} ||_{\lambda \in \Lambda, Lip} L^{\hat{\theta}} + L_2^\ell || \frac{\partial}{\partial \lambda} \hat{\theta}(\lambda, S^{tr}) ||_{\lambda \in \Lambda, Lip}$$

$$\leq \gamma_1^\ell + \gamma_4^\ell L^{\hat{\theta}} + (\gamma_3^\ell + \gamma_2^\ell L^{\hat{\theta}}) L^{\hat{\theta}} + L_2^\ell \gamma^{\hat{\theta}} \triangleq \gamma. \tag{5}$$

With Eq. (4) and Eq. (5), we can conclude $\ell(\lambda, \hat{\theta}(\lambda, S^{tr}), z)$ as a function of $\lambda$ is $L = \mathcal{O}((1 + \eta(\gamma_\varphi - \nu))^K)$ Lipschitz continuous and $\gamma = \mathcal{O}((1 + \eta(\gamma_\varphi - \nu))^{2K})$ Lipschitz smooth. By definition, $L$ and $\gamma$ are independent of the training dataset $S^{tr}$, $z$, the random indices $(j_1, j_2, \cdots, j_K)$ and thereby the randomness of $\hat{\theta}$. Thereby, we have $\forall S^{tr} \in Z^n$, $\forall z \in Z$, $\forall g \in \mathcal{G}_{\hat{\theta}}$, $\ell(\lambda, g(\lambda, S^{tr}), z)$ as a function of $\lambda$ is $L = \mathcal{O}((1 + \eta(\gamma_\varphi - \nu))^K)$ Lipschitz continuous and $\gamma = \mathcal{O}((1 + \eta(\gamma_\varphi - \nu))^{2K})$ Lipschitz smooth. Similarly, the result also holds for GD. $\qquad \square$

### A.4  Proof of Theorem 4

**Theorem 4** (Expectation bound of CV). *Suppose $S^{tr} \sim (D^{tr})^n$, $S^{val} \sim (D^{val})^m$ and $S^{tr}$ and $S^{val}$ are independent, and let $\mathbf{A}^{cv}(S^{tr}, S^{val})$ denote the results of CV as shown in Algorithm 2, then*

$$|\mathbb{E} \left[ R(\mathbf{A}^{cv}(S^{tr}, S^{val}), D^{val}) - \hat{R}^{val}(\mathbf{A}^{cv}(S^{tr}, S^{val}), S^{val}) \right] | \leq s(\ell) \sqrt{\frac{\log T}{2m}}.$$

*Proof.* Let $\lambda_t \in \Lambda$ be the $t$th hyperparameter, which is a random vector taking value on $\Lambda$, $\hat{\theta}^t$ be the random function corresponding to the $t$th optimization in the inner level, then $\hat{\theta}^t(\lambda_t, S^{tr})$ is the output hypothesis given hyperparameter $\lambda_t$ and training dataset $S^{tr}$. Let $t^*$ be the index of the best hyperparameter, i.e.,

$$t^* = \arg\min_{1 \leq t \leq T} \hat{R}^{val}(\lambda_t, \hat{\theta}^t(\lambda_t, S^{tr}), S^{val}),$$

then the output of CV is $\mathbf{A}^{cv}(S^{tr}, S^{val}) = (\lambda_{t^*}, \hat{\theta}^{t^*}(\lambda_{t^*}, S^{tr}))$.

Let $X_t = R(\lambda_t, \hat{\theta}^t(\lambda_t, S^{tr}), D^{val}) - \hat{R}^{val}(\lambda_t, \hat{\theta}^t(\lambda_t, S^{tr}), S^{val})$, then we have

$$R(\mathbf{A}^{cv}(S^{tr}, S^{val}), D^{val}) - \hat{R}^{val}(\mathbf{A}^{cv}(S^{tr}, S^{val}), S^{val})$$

$$= R(\lambda_{t^*}, \hat{\theta}^{t^*}(\lambda_{t^*}, S^{tr}), D^{val}) - \hat{R}^{val}(\lambda_{t^*}, \hat{\theta}^{t^*}(\lambda_{t^*}, S^{tr}), S^{val}) = X_{t^*}.$$

By Hoeffding's lemma, we have for any $s > 0$

$$\mathbf{E} e^{sX_t} = \mathbf{E}_{\lambda_t, \hat{\theta}^t, S^{tr}} \mathbf{E}_{S^{val}} \exp \left( \frac{s}{m} \sum_{k=1}^m R(\lambda_t, \hat{\theta}^t(\lambda_t, S^{tr}), D^{val}) - \ell(\lambda_t, \hat{\theta}^t(\lambda_t, S^{tr}), z_k^{val}) \right)$$

$$= \mathbf{E}_{\lambda_t, \hat{\theta}^t, S^{tr}} \prod_{k=1}^m \mathbf{E}_{z_k^{val}} \exp \left( \frac{s}{m} \left( R(\lambda_t, \hat{\theta}^t(\lambda_t, S^{tr}), D^{val}) - \ell(\lambda_t, \hat{\theta}^t(\lambda_t, S^{tr}), z_k^{val}) \right) \right)$$

$$\leq \prod_{k=1}^m \exp(\frac{s^2}{m^2} \frac{s(\ell)^2}{8}) = \exp(\frac{s^2}{m} \frac{s(\ell)^2}{8}).$$

Then we have

$$\mathbf{E}X_{t^*} \leq \mathbf{E} \max_{1 \leq t \leq T} X_t = \frac{1}{s}\mathbf{E}\log\exp(s\max_{1 \leq t \leq T} X_t) \leq \frac{1}{s}\log\mathbf{E}\exp(s\max_{1 \leq t \leq T} X_t)$$

$$= \frac{1}{s}\log\mathbf{E}\max_{1 \leq t \leq T}\exp(sX_t) \leq \frac{1}{s}\log\sum_{1 \leq t \leq T}\mathbf{E}\exp(sX_t)$$

$$\leq \frac{1}{s}\log\left(T\exp(\frac{s^2}{m}\frac{s(\ell)^2}{8})\right) = \frac{\log T}{s} + \frac{s \cdot s(\ell)^2}{8m}.$$

Taking $s = \sqrt{\frac{8m\log T}{s(\ell)^2}}$, we have $\mathbf{E}X_{t^*} \leq s(\ell)\sqrt{\frac{\log T}{2m}}$. Similarly, we have $-\mathbf{E}X_{t^*} \leq s(\ell)\sqrt{\frac{\log T}{2m}}$.

Finally, $|\mathbf{E}X_{t^*}| \leq s(\ell)\sqrt{\frac{\log T}{2m}}$. $\qquad\qquad\qquad\qquad\qquad\qquad\qquad\qquad\qquad\qquad\square$

## B  Construct a Worst Case for Theorem 3

We construct a worst case where the Lipschitz constant $L$ in Theorem 3 increases at least exponentially w.r.t. $K$. It is a feature learning example with a small neural network. The model has one parameter and one hyperparameter and uses squared activation function [1]. We use the squared loss. The data distribution is any distribution in the support $Z = \{(x,y) : \frac{1}{2} \leq x \leq 1, 1 \leq y \leq 2\}$. The parameter space and hyperparameter space are $\Theta = [0,1]$ and $\Lambda = [0, \frac{1}{4}]$ respectively. Formally, the loss function is $\ell(\lambda, \theta, z) = (y - \lambda(\theta x)^2)^2$. The inner loop is solved by SGD with a learning rate $\eta$. We formalize the result in Proposition 1.

**Proposition 1.** *Suppose* $\ell(\lambda, \theta, z) = (y - \lambda(\theta x)^2)^2$, $\Lambda = [0, \frac{1}{4}]$, $\Theta = [0,1]$, $Z = \{(x,y) : \frac{1}{2} \leq x \leq 1, 1 \leq y \leq 2\}$ *and the inner level problem is solved with $K$ steps SGD with learning rate $\eta$, then* $\forall S^{tr} \in Z^n$, $\forall z \in Z$, $\forall g \in \mathcal{G}_{\hat{\theta}}$, $\ell(\lambda, g(\lambda, S^{tr}), z)$ *as a function of $\lambda$ is at least $L = \Omega((1 + \frac{3}{16}\eta)^K)$ Lipschitz continuous.*

*Proof.* We use $z = (x,y) \in Z$ to denote the data point used in one step of SGD, where we omit the index of the data point for simplicity. Firstly, the gradient of the loss function is $\nabla_\theta \ell(\lambda, \theta, z) = 2(y - \lambda(\theta x)^2)(-\lambda x^2 2\theta) = -4(y\lambda x^2\theta - \lambda^2\theta^3 x^4)$ and one step SGD satisfies

$$\theta - \eta\nabla_\theta\ell(\lambda,\theta,z) = \theta + 4\eta(y\lambda x^2\theta - \lambda^2\theta^3 x^4) = (1 + 4\eta y\lambda x^2)\theta - 4\eta\lambda^2\theta^3 x^4$$

$$\geq (1 + 4\eta y\lambda x^2)\theta - 4\eta\lambda^2\theta x^4 = (1 + 4\eta y\lambda x^2 - 4\eta\lambda^2 x^4)\theta \geq (1 + 3\eta\lambda x^2)\theta \geq (1 + \frac{3}{4}\eta\lambda)\theta.$$

Let $\{\hat{\theta}_k(\lambda)\}_{k \geq 0}$ be the trajectory of SGD, then we have $\hat{\theta}_k(\lambda) \geq (1 + \frac{3}{4}\eta\lambda)^k\theta_0$.

Taking gradient of $\hat{\theta}_k(\lambda)$ w.r.t. $\lambda$, we have

$$\nabla_\lambda\hat{\theta}_{k+1}(\lambda) = 4\eta yx^2\hat{\theta}_k(\lambda) + (1 + 4\eta y\lambda x^2)\nabla_\lambda\hat{\theta}_k(\lambda) - 4\eta x^4(2\lambda\hat{\theta}_k(\lambda)^3 + \lambda^2 3\hat{\theta}_k(\lambda)^2\nabla_\lambda\hat{\theta}_k(\lambda))$$

$$= 4\eta yx^2\hat{\theta}_k(\lambda) + (1 + 4\eta y\lambda x^2)\nabla_\lambda\hat{\theta}_k(\lambda) - 8\eta x^4\lambda\hat{\theta}_k(\lambda)^3 - 12\eta x^4\lambda^2\hat{\theta}_k(\lambda)^2\nabla_\lambda\hat{\theta}_k(\lambda)$$

$$= 4\eta x^2\hat{\theta}_k(\lambda)(y - 2x^2\lambda\hat{\theta}_k(\lambda)^2) + (1 + 4\eta y\lambda x^2 - 12\eta x^4\lambda^2\hat{\theta}_k(\lambda)^2)\nabla_\lambda\hat{\theta}_k(\lambda)$$

As for the first term, we have $4\eta x^2\hat{\theta}_k(\lambda)(y - 2x^2\lambda\hat{\theta}_k(\lambda)^2) \geq 2\eta x^2\hat{\theta}_k(\lambda) \geq 0$. As for the coefficient of the second term, we have $1 + 4\eta y\lambda x^2 - 12\eta x^4\lambda^2\hat{\theta}_k(\lambda)^2 \geq 1 + \eta\lambda x^2 \geq 1 + \eta\lambda/4 \geq 0$. Besides, $\nabla_\lambda\hat{\theta}_1(\lambda) = 4\eta yx^2\theta_0 - 8\eta\lambda x^4\theta_0^3 \geq 2x^2\theta_0\eta \geq \frac{1}{2}\theta_0\eta$. Thereby, $\nabla_\lambda\hat{\theta}_k(\lambda) \geq 0$ and furthermore

$$\nabla_\lambda\hat{\theta}_{k+1}(\lambda) \geq (1 + \eta\lambda/4)\nabla_\lambda\hat{\theta}_k(\lambda) \geq (1 + \eta\lambda/4)^k\nabla_\lambda\hat{\theta}_1(\lambda) \geq \frac{1}{2}(1 + \eta\lambda/4)^k\theta_0\eta.$$

Then, we consider $\ell(\lambda, \hat{\theta}_K(\lambda), z) = (y - \lambda(\hat{\theta}_K(\lambda)x)^2)^2$. Its gradient w.r.t. $\lambda$ is

$$\nabla_\lambda\ell(\lambda, \hat{\theta}_K(\lambda), z) = 2(y - \lambda(\hat{\theta}_K(\lambda)x)^2)(-(\hat{\theta}_K(\lambda)x)^2 - 2\lambda x^2\hat{\theta}_K(\lambda)\nabla_\lambda\hat{\theta}_K(\lambda)).$$

Thereby,

$$|\nabla_\lambda\ell(\lambda, \hat{\theta}_K(\lambda), z)| = 2|y - \lambda(\hat{\theta}_K(\lambda)x)^2| \cdot |(\hat{\theta}_K(\lambda)x)^2 + 2\lambda x^2\hat{\theta}_K(\lambda)\nabla_\lambda\hat{\theta}_K(\lambda)|$$

$$= 2|y - \lambda(\hat{\theta}_K(\lambda)x)^2| \cdot |\hat{\theta}_K(\lambda) + 2\lambda\nabla_\lambda\hat{\theta}_K(\lambda)| \cdot \hat{\theta}_K(\lambda) \cdot x^2.$$

Since $|y - \lambda(\hat{\theta}_K(\lambda)x)^2| \geq (1 - \frac{1}{4}) = \frac{3}{4}$, $|\hat{\theta}_K(\lambda) + 2\lambda\nabla_\lambda\hat{\theta}_K(\lambda)| \geq \lambda(1 + \eta\lambda/4)^{K-1}\theta_0\eta$ and $\hat{\theta}_K(\lambda) \geq (1 + \frac{3}{4}\eta\lambda)^K\theta_0$, we have

$$|\nabla_\lambda\ell(\lambda, \hat{\theta}_K(\lambda), z)| \geq 2 \cdot \frac{3}{4} \cdot \lambda(1 + \eta\lambda/4)^{K-1}\theta_0\eta \cdot (1 + \frac{3}{4}\eta\lambda)^K\theta_0 \cdot \frac{1}{4}$$
$$= \frac{3}{8}\lambda(1 + \eta\lambda/4)^{K-1}\theta_0^2\eta(1 + 3\eta\lambda/4)^K.$$

Finally,

$$||\ell(\lambda, \hat{\theta}(\lambda), z)||_{Lip} \geq \sup_{\lambda\in\Lambda}\frac{3}{8}\lambda(1 + \eta\lambda/4)^{K-1}\theta_0^2\eta(1 + 3\eta\lambda/4)^K$$
$$\geq \frac{3}{32}(1 + \eta/16)^{K-1}\theta_0^2\eta(1 + 3\eta/16)^K := L,$$

and $||\ell(\lambda, \hat{\theta}(\lambda), z)||_{Lip} \geq L = \Omega((1 + \frac{3}{16}\eta)^K)$. $\qquad\square$

## C   Improve Theorem 3 under Stronger Assumptions

When the inner loss $\varphi_i$ is convex or strongly convex, we can get tighter bounds for $L$ and $\gamma$ in Theorem 3. In Proposition 2, we show that $L = \mathcal{O}(K)$ and $\gamma = \mathcal{O}(K^3)$ when the inner loss $\varphi_i$ is convex. In this case, the dependence on $K$ of the generalization gap (i.e., $\beta$ in Theorem 2) is $\mathcal{O}(K^2)$. In Proposition 3, we show that $L = \mathcal{O}(1)$ and $\gamma = \mathcal{O}(1)$ w.r.t. $K$ when the inner loss $\varphi_i$ is strongly convex. In this case, the dependence on $K$ of the generalization gap is $\mathcal{O}(1)$. We get these tighter results by deriving tighter Lipschitz constants for updating functions of SGD w.r.t. $\theta$, using the (strongly) convex properties of $\varphi_i$. Other parts of the proof is the same as Theorem 3.

Notice that Theorem 3 implies that the learning rate $\eta$ in the inner level should be of the order of $1/K$ for a moderate $L$ and $\gamma$. Therefore, $\eta$ will be very small when $K$ is very large, and the algorithm will converge slow in practice. However, Proposition 2 and Proposition 3 imply that if we use a (strongly) convex inner loss, $\eta$ will not affect the order of $L$ and $\gamma$, and thereby we can use a larger $\eta$ in practice in this case.

**Proposition 2.** *Suppose Assumption 1,2,3,4 hold, $\varphi_i(\lambda, \theta, z)$ as a function of $\theta$ is convex for all $1 \leq i \leq n$, $z \in Z$ and $\lambda \in \Lambda$, and the inner level problem is solved with $K$ steps SGD or GD with learning rate $\eta \leq \frac{2}{\gamma_\varphi}$, then $\forall S^{tr} \in Z^n$, $\forall z \in Z$, $\forall g \in \mathcal{G}_{\hat{\theta}}$, $\ell(\lambda, g(\lambda, S^{tr}), z)$ as a function of $\lambda$ is $L = \mathcal{O}(K)$ Lipschitz continuous and $\gamma = \mathcal{O}(K^3)$ Lipschitz smooth.*

*Proof.* The $k$th updating step of SGD can be written as

$$G_{\lambda,k}(\theta) = \theta - \eta\nabla_\theta\varphi_{j_k}(\lambda, \theta, z_{j_k}^{tr}) = \nabla_\theta\left(\frac{1}{2}||\theta||^2 - \eta\varphi_{j_k}(\lambda, \theta, z_{j_k}^{tr})\right),$$

where $j_k$ is randomly selected from $\{1, 2, \cdots, n\}$. The output of $K$ steps SGD is $\hat{\theta}(\lambda, S^{tr}) = G_{\lambda,K}(G_{\lambda,K-1}(\cdots(G_{\lambda,1}(\theta_0))))$ and $\mathcal{G}_{\hat{\theta}}$ is formed by iterates over $(j_1, j_2, \cdots, j_K) \in \{1, 2, \cdots, n\}^K$.

According to Lemma 2 and Assumption 3, we have

$$L_1^G \triangleq \sup_{k, j_k, S^{tr}, \theta}||G_{\lambda,k}(\theta)||_{\lambda\in\Lambda, Lip} = \sup_{i,z,\theta}||\nabla_\theta\left(\frac{1}{2}||\theta||^2 - \eta\varphi_i(\lambda, \theta, z)\right)||_{\lambda\in\Lambda, Lip} < \infty.$$

Similarly, we have

$$\gamma_1^G \triangleq \sup_{k, j_k, S^{tr}, \theta}||\frac{\partial}{\partial\lambda}G_{\lambda,k}(\theta)||_{\lambda\in\Lambda, Lip} < \infty, \quad \gamma_2^G \triangleq \sup_{k, j_k, S^{tr}, \lambda}||\frac{\partial}{\partial\theta}G_{\lambda,k}(\theta)||_{\theta\in\Theta, Lip} < \infty,$$

$$\gamma_3^G \triangleq \sup_{k, j_k, S^{tr}, \theta}||\frac{\partial}{\partial\theta}G_{\lambda,k}(\theta)||_{\lambda\in\Lambda, Lip} < \infty, \quad \gamma_4^G \triangleq \sup_{k, j_k, S^{tr}, \lambda}||\frac{\partial}{\partial\lambda}G_{\lambda,k}(\theta)||_{\theta\in\Theta, Lip} < \infty.$$

Then we consider $||G_{\lambda,k}(\theta)||_{\theta\in\Theta,Lip}$. According to the co-coercivity of $\nabla_\theta\varphi_{j_k}(\lambda,\theta,z_{j_k}^{tr})$, we have

$$
\begin{aligned}
||G_{\lambda,k}(\theta)-G_{\lambda,k}(\theta')||^2 =&||\theta-\theta'||^2 + \eta^2||\nabla_\theta\varphi_{j_k}(\lambda,\theta,z_{j_k}^{tr})-\nabla_\theta\varphi_{j_k}(\lambda,\theta',z_{j_k}^{tr})||^2\\
&-2\eta\left\langle\theta-\theta',\nabla_\theta\varphi_{j_k}(\lambda,\theta,z_{j_k}^{tr})-\nabla_\theta\varphi_{j_k}(\lambda,\theta',z_{j_k}^{tr})\right\rangle\\
\leq&||\theta-\theta'||^2 + \eta^2||\nabla_\theta\varphi_{j_k}(\lambda,\theta,z_{j_k}^{tr})-\nabla_\theta\varphi_{j_k}(\lambda,\theta',z_{j_k}^{tr})||^2\\
&-2\frac{\eta}{\gamma_\varphi}||\nabla_\theta\varphi_{j_k}(\lambda,\theta,z_{j_k}^{tr})-\nabla_\theta\varphi_{j_k}(\lambda,\theta',z_{j_k}^{tr})||^2 \leq ||\theta-\theta'||^2.
\end{aligned}
$$

Thereby, $||G_{\lambda,k}(\theta)||_{\theta\in\Theta,Lip} \leq 1$ and $\sup_{k,j_k,S^{tr},\lambda}||G_{\lambda,k}(\theta)||_{\theta\in\Theta,Lip} \leq 1 \triangleq L_2^G$. According to Lemma 3 and Lemma 4, $\hat\theta(\lambda,S^{tr})$ is $L^{\hat\theta} = KL_1^G$ Lipschitz continuous and $\gamma^{\hat\theta} = \mathcal{O}(K^3)$ Lipschitz smooth as a function of $\lambda$. By definition, $L^{\hat\theta}$ and $\gamma^{\hat\theta}$ are independent of the training dataset $S^{tr}$ and the random indices $(j_1,j_2,\cdots,j_K)$ and thereby the randomness of $\hat\theta$.

According to Lemma 2 and Assumption 2, we have

$$
L_1^\ell = \sup_{\theta\in\Theta,z\in Z}||\ell(\lambda,\theta,z)||_{\lambda\in\Lambda,Lip} < \infty,\ L_2^\ell = \sup_{\lambda\in\Lambda,z\in Z}||\ell(\lambda,\theta,z)||_{\theta\in\Theta,Lip} < \infty.
$$

Similarly, we have

$$
\gamma_1^\ell \triangleq \sup_{\theta,z}||\left[\frac{\partial}{\partial\lambda}\ell(\lambda,\theta,z)\right]||_{\lambda\in\Lambda,Lip} < \infty,\ \gamma_2^\ell \triangleq \sup_{\lambda,z}||\left[\frac{\partial}{\partial\theta}\ell(\lambda,\theta,z)\right]||_{\theta\in\Theta,Lip} < \infty,
$$

$$
\gamma_3^\ell \triangleq \sup_{\theta,z}||\left[\frac{\partial}{\partial\theta}\ell(\lambda,\theta,z)\right]||_{\lambda\in\Lambda,Lip} < \infty,\ \gamma_4^\ell \triangleq \sup_{\lambda,z}||\left[\frac{\partial}{\partial\lambda}\ell(\lambda,\theta,z)\right]||_{\theta\in\Theta,Lip} < \infty.
$$

Suppose $z\in Z$, firstly we consider the Lipschitz continuity of $\ell(\lambda,\hat\theta(\lambda,S^{tr}),z)$:

$$
\begin{aligned}
&||\ell(\lambda,\hat\theta(\lambda,S^{tr}),z)||_{\lambda\in\Lambda,Lip}\\
\leq& \sup_{\theta\in\Theta,z\in Z}||\ell(\lambda,\theta,z)||_{\lambda\in\Lambda,Lip} + \sup_{\lambda\in\Lambda,z\in Z}||\ell(\lambda,\theta,z)||_{\theta\in\Theta,Lip}\cdot||\hat\theta(\lambda,S^{tr})||_{\lambda\in\Lambda,Lip}\\
\leq& L_1^\ell + L_2^\ell L^{\hat\theta} \triangleq L.
\end{aligned} \tag{6}
$$

Then we consider the Lipschitz continuity of $\frac{\partial}{\partial\lambda}\ell(\lambda,\hat\theta(\lambda,S^{tr}),z)$, which can be expanded as

$$
\frac{\partial}{\partial\lambda}\ell(\lambda,\hat\theta(\lambda,S^{tr}),z) = \left[\frac{\partial}{\partial\lambda}\ell(\lambda,\theta,z)\right]\Big|_{\theta=\hat\theta(\lambda,S^{tr})} + \left[\frac{\partial}{\partial\theta}\ell(\lambda,\theta,z)\right]\Big|_{\theta=\hat\theta(\lambda,S^{tr})}\left[\frac{\partial}{\partial\lambda}\hat\theta(\lambda,S^{tr})\right].
$$

Taking the Lipschitz norm w.r.t. $\lambda$, we have

$$
||\left[\frac{\partial}{\partial\lambda}\ell(\lambda,\theta,z)\right]\Big|_{\theta=\hat\theta(\lambda,S^{tr})}||_{\lambda\in\Lambda,Lip} \leq \gamma_1^\ell + \gamma_4^\ell L^{\hat\theta},
$$

$$
||\left[\frac{\partial}{\partial\theta}\ell(\lambda,\theta,z)\right]\Big|_{\theta=\hat\theta(\lambda,S^{tr})}||_{\lambda\in\Lambda,Lip} \leq \gamma_3^\ell + \gamma_2^\ell L^{\hat\theta},
$$

which yields

$$
\begin{aligned}
&||\frac{\partial}{\partial\lambda}\ell(\lambda,\hat\theta(\lambda,S^{tr}),z)||_{\lambda\in\Lambda,Lip}\\
\leq&||\left[\frac{\partial}{\partial\lambda}\ell(\lambda,\theta,z)\right]\Big|_{\theta=\hat\theta(\lambda,S^{tr})}||_{\lambda\in\Lambda,Lip}\\
&+||\left[\frac{\partial}{\partial\theta}\ell(\lambda,\theta,z)\right]\Big|_{\theta=\hat\theta(\lambda,S^{tr})}||_{\lambda\in\Lambda,Lip}L^{\hat\theta} + L_2^\ell||\frac{\partial}{\partial\lambda}\hat\theta(\lambda,S^{tr})||_{\lambda\in\Lambda,Lip}\\
\leq&\gamma_1^\ell + \gamma_4^\ell L^{\hat\theta} + (\gamma_3^\ell + \gamma_2^\ell L^{\hat\theta})L^{\hat\theta} + L_2^\ell\gamma^{\hat\theta} \triangleq \gamma.
\end{aligned} \tag{7}
$$

With Eq. (6) and Eq. (7), we can conclude $\ell(\lambda, \hat{\theta}(\lambda, S^{tr}), z)$ as a function of $\lambda$ is $L = \mathcal{O}(K)$ Lipschitz continuous and $\gamma = \mathcal{O}(K^3)$ Lipschitz smooth. By definition, $L$ and $\gamma$ are independent of the training dataset $S^{tr}$, $z$, the random indices $(j_1, j_2, \cdots, j_K)$ and thereby the randomness of $\hat{\theta}$. Thereby, we have $\forall S^{tr} \in Z^n$, $\forall z \in Z$, $\forall g \in \mathcal{G}_{\hat{\theta}}$, $\ell(\lambda, g(\lambda, S^{tr}), z)$ as a function of $\lambda$ is $L = \mathcal{O}(K)$ Lipschitz continuous and $\gamma = \mathcal{O}(K^3)$ Lipschitz smooth. Similarly, the result also holds for GD. □

**Proposition 3.** *Suppose Assumption 1,2,3,4 hold, $\varphi_i(\lambda, \theta, z)$ as a function of $\theta$ is $\tau$-strongly convex for all $1 \leq i \leq n$, $z \in Z$ and $\lambda \in \Lambda$, and the inner level problem is solved with $K$ steps SGD or GD with learning rate $\eta \leq \frac{1}{\gamma_\varphi}$, then $\forall S^{tr} \in Z^n$, $\forall z \in Z$, $\forall g \in \mathcal{G}_{\hat{\theta}}$, $\ell(\lambda, g(\lambda, S^{tr}), z)$ as a function of $\lambda$ is $L = \mathcal{O}(1)$ Lipschitz continuous and $\gamma = \mathcal{O}(1)$ Lipschitz smooth w.r.t. $K$.*

*Proof.* The $k$th updating step of SGD can be written as

$$G_{\lambda,k}(\theta) = \theta - \eta \nabla_\theta \varphi_{j_k}(\lambda, \theta, z_{j_k}^{tr}) = \nabla_\theta \left( \frac{1}{2} ||\theta||^2 - \eta \varphi_{j_k}(\lambda, \theta, z_{j_k}^{tr}) \right),$$

where $j_k$ is randomly selected from $\{1, 2, \cdots, n\}$. The output of $K$ steps SGD is $\hat{\theta}(\lambda, S^{tr}) = G_{\lambda,K}(G_{\lambda,K-1}(\cdots(G_{\lambda,1}(\theta_0))))$ and $\mathcal{G}_{\hat{\theta}}$ is formed by iterates over $(j_1, j_2, \cdots, j_K) \in \{1, 2, \cdots, n\}^K$.

According to Lemma 2 and Assumption 3, we have

$$L_1^G \triangleq \sup_{k,j_k,S^{tr},\theta} ||G_{\lambda,k}(\theta)||_{\lambda \in \Lambda, Lip} = \sup_{i,z,\theta} ||\nabla_\theta \left( \frac{1}{2} ||\theta||^2 - \eta \varphi_i(\lambda, \theta, z) \right) ||_{\lambda \in \Lambda, Lip} < \infty.$$

Similarly, we have

$$\gamma_1^G \triangleq \sup_{k,j_k,S^{tr},\theta} ||\frac{\partial}{\partial \lambda} G_{\lambda,k}(\theta)||_{\lambda \in \Lambda, Lip} < \infty, \quad \gamma_2^G \triangleq \sup_{k,j_k,S^{tr},\lambda} ||\frac{\partial}{\partial \theta} G_{\lambda,k}(\theta)||_{\theta \in \Theta, Lip} < \infty,$$

$$\gamma_3^G \triangleq \sup_{k,j_k,S^{tr},\theta} ||\frac{\partial}{\partial \theta} G_{\lambda,k}(\theta)||_{\lambda \in \Lambda, Lip} < \infty, \quad \gamma_4^G \triangleq \sup_{k,j_k,S^{tr},\lambda} ||\frac{\partial}{\partial \lambda} G_{\lambda,k}(\theta)||_{\theta \in \Theta, Lip} < \infty.$$

Then we consider $||G_{\lambda,k}(\theta)||_{\theta \in \Theta, Lip}$. Since $\varphi_{j_k}(\lambda, \theta, z_{j_k}^{tr})$ as a function of $\theta$ is $\tau$-strongly convex, we have $\varphi_{j_k}(\lambda, \theta, z_{j_k}^{tr}) - \frac{\tau}{2} ||\theta||^2$ as a function of $\theta$ is convex and $\gamma_\varphi - \tau$ Lipschitz smooth. According to the co-coercivity of $\nabla_\theta(\varphi_{j_k}(\lambda, \theta, z_{j_k}^{tr}) - \frac{\tau}{2} ||\theta||^2)$, we have

$$\left\langle \theta - \theta', \nabla_\theta \varphi_{j_k}(\lambda, \theta, z_{j_k}^{tr}) - \tau\theta - \nabla_\theta \varphi_{j_k}(\lambda, \theta', z_{j_k}^{tr}) + \tau\theta' \right\rangle$$
$$\geq \frac{1}{\gamma_\varphi - \tau} ||\nabla_\theta \varphi_{j_k}(\lambda, \theta, z_{j_k}^{tr}) - \tau\theta - \nabla_\theta \varphi_{j_k}(\lambda, \theta', z_{j_k}^{tr}) + \tau\theta'||^2,$$

which is equivalent to

$$\left\langle \theta - \theta', \nabla_\theta \varphi_{j_k}(\lambda, \theta, z_{j_k}^{tr}) - \nabla_\theta \varphi_{j_k}(\lambda, \theta', z_{j_k}^{tr}) \right\rangle$$
$$\geq \frac{1}{\gamma_\varphi + \tau} ||\nabla_\theta \varphi_{j_k}(\lambda, \theta, z_{j_k}^{tr}) - \nabla_\theta \varphi_{j_k}(\lambda, \theta', z_{j_k}^{tr})||^2 + \frac{\gamma_\varphi \tau}{\gamma_\varphi + \tau} ||\theta - \theta'||^2.$$

As a result,

$$||G_{\lambda,k}(\theta) - G_{\lambda,k}(\theta')||^2$$
$$= ||\theta - \theta'||^2 + \eta^2 ||\nabla_\theta \varphi_{j_k}(\lambda, \theta, z_{j_k}^{tr}) - \nabla_\theta \varphi_{j_k}(\lambda, \theta', z_{j_k}^{tr})||^2$$
$$\quad - 2\eta \left\langle \theta - \theta', \nabla_\theta \varphi_{j_k}(\lambda, \theta, z_{j_k}^{tr}) - \nabla_\theta \varphi_{j_k}(\lambda, \theta', z_{j_k}^{tr}) \right\rangle$$
$$\leq ||\theta - \theta'||^2 + \eta^2 ||\nabla_\theta \varphi_{j_k}(\lambda, \theta, z_{j_k}^{tr}) - \nabla_\theta \varphi_{j_k}(\lambda, \theta', z_{j_k}^{tr})||^2$$
$$\quad - 2\eta(\frac{1}{\gamma_\varphi + \tau} ||\nabla_\theta \varphi_{j_k}(\lambda, \theta, z_{j_k}^{tr}) - \nabla_\theta \varphi_{j_k}(\lambda, \theta', z_{j_k}^{tr})||^2 + \frac{\gamma_\varphi \tau}{\gamma_\varphi + \tau} ||\theta - \theta'||^2)$$
$$= (1 - 2\eta \frac{\gamma_\varphi \tau}{\gamma_\varphi + \tau}) ||\theta - \theta'||^2 + (\eta^2 - \frac{2\eta}{\gamma_\varphi + \tau}) ||\nabla_\theta \varphi_{j_k}(\lambda, \theta, z_{j_k}^{tr}) - \nabla_\theta \varphi_{j_k}(\lambda, \theta', z_{j_k}^{tr})||^2.$$

Since $\eta \leq \frac{1}{\gamma_\varphi} \leq \frac{2}{\gamma_\varphi+\tau}$, we have $||G_{\lambda,k}(\theta)||_{\theta\in\Theta,Lip} \leq \sqrt{1-2\eta\frac{\gamma_\varphi\tau}{\gamma_\varphi+\tau}}$ and

$$\sup_{k,j_k,S^{tr},\lambda} ||G_{\lambda,k}(\theta)||_{\theta\in\Theta,Lip} \leq \sqrt{1-2\eta\frac{\gamma_\varphi\tau}{\gamma_\varphi+\tau}} \triangleq L_2^G < 1.$$

According to Lemma 3 and Lemma 4, $\hat{\theta}(\lambda, S^{tr})$ as a function of $\lambda$ is $L^{\hat{\theta}} = \mathcal{O}(1)$ Lipschitz continuous and $\gamma^{\hat{\theta}} = \mathcal{O}(1)$ Lipschitz smooth w.r.t. $K$. By definition, $L^{\hat{\theta}}$ and $\gamma^{\hat{\theta}}$ are independent of the training dataset $S^{tr}$ and the random indices $(j_1, j_2, \cdots, j_K)$ and thereby the randomness of $\hat{\theta}$.

According to Lemma 2 and Assumption 2, we have

$$L_1^\ell = \sup_{\theta\in\Theta,z\in Z} ||\ell(\lambda,\theta,z)||_{\lambda\in\Lambda,Lip} < \infty, \ L_2^\ell = \sup_{\lambda\in\Lambda,z\in Z} ||\ell(\lambda,\theta,z)||_{\theta\in\Theta,Lip} < \infty.$$

Similarly, we have

$$\gamma_1^\ell \triangleq \sup_{\theta,z} ||\left[\frac{\partial}{\partial\lambda}\ell(\lambda,\theta,z)\right]||_{\lambda\in\Lambda,Lip} < \infty, \ \gamma_2^\ell \triangleq \sup_{\lambda,z} ||\left[\frac{\partial}{\partial\theta}\ell(\lambda,\theta,z)\right]||_{\theta\in\Theta,Lip} < \infty,$$

$$\gamma_3^\ell \triangleq \sup_{\theta,z} ||\left[\frac{\partial}{\partial\theta}\ell(\lambda,\theta,z)\right]||_{\lambda\in\Lambda,Lip} < \infty, \ \gamma_4^\ell \triangleq \sup_{\lambda,z} ||\left[\frac{\partial}{\partial\lambda}\ell(\lambda,\theta,z)\right]||_{\theta\in\Theta,Lip} < \infty.$$

Suppose $z \in Z$, firstly we consider the Lipschitz continuity of $\ell(\lambda, \hat{\theta}(\lambda, S^{tr}), z)$:

$$||\ell(\lambda, \hat{\theta}(\lambda, S^{tr}), z)||_{\lambda\in\Lambda,Lip}$$
$$\leq \sup_{\theta\in\Theta,z\in Z} ||\ell(\lambda,\theta,z)||_{\lambda\in\Lambda,Lip} + \sup_{\lambda\in\Lambda,z\in Z} ||\ell(\lambda,\theta,z)||_{\theta\in\Theta,Lip} \cdot ||\hat{\theta}(\lambda,S^{tr})||_{\lambda\in\Lambda,Lip}$$
$$\leq L_1^\ell + L_2^\ell L^{\hat{\theta}} \triangleq L. \tag{8}$$

Then we consider the Lipschitz continuity of $\frac{\partial}{\partial\lambda}\ell(\lambda, \hat{\theta}(\lambda, S^{tr}), z)$, which can be expanded as

$$\frac{\partial}{\partial\lambda}\ell(\lambda, \hat{\theta}(\lambda, S^{tr}), z) = \left[\frac{\partial}{\partial\lambda}\ell(\lambda,\theta,z)\right]\Big|_{\theta=\hat{\theta}(\lambda,S^{tr})} + \left[\frac{\partial}{\partial\theta}\ell(\lambda,\theta,z)\right]\Big|_{\theta=\hat{\theta}(\lambda,S^{tr})}\left[\frac{\partial}{\partial\lambda}\hat{\theta}(\lambda, S^{tr})\right].$$

Taking the Lipschitz norm w.r.t. $\lambda$, we have

$$||\left[\frac{\partial}{\partial\lambda}\ell(\lambda,\theta,z)\right]\Big|_{\theta=\hat{\theta}(\lambda,S^{tr})}||_{\lambda\in\Lambda,Lip} \leq \gamma_1^\ell + \gamma_4^\ell L^{\hat{\theta}},$$

$$||\left[\frac{\partial}{\partial\theta}\ell(\lambda,\theta,z)\right]\Big|_{\theta=\hat{\theta}(\lambda,S^{tr})}||_{\lambda\in\Lambda,Lip} \leq \gamma_3^\ell + \gamma_2^\ell L^{\hat{\theta}},$$

which yields

$$||\frac{\partial}{\partial\lambda}\ell(\lambda, \hat{\theta}(\lambda, S^{tr}), z)||_{\lambda\in\Lambda,Lip}$$
$$\leq ||\left[\frac{\partial}{\partial\lambda}\ell(\lambda,\theta,z)\right]\Big|_{\theta=\hat{\theta}(\lambda,S^{tr})}||_{\lambda\in\Lambda,Lip}$$
$$+ ||\left[\frac{\partial}{\partial\theta}\ell(\lambda,\theta,z)\right]\Big|_{\theta=\hat{\theta}(\lambda,S^{tr})}||_{\lambda\in\Lambda,Lip}L^{\hat{\theta}} + L_2^\ell||\frac{\partial}{\partial\lambda}\hat{\theta}(\lambda, S^{tr})||_{\lambda\in\Lambda,Lip}$$
$$\leq \gamma_1^\ell + \gamma_4^\ell L^{\hat{\theta}} + (\gamma_3^\ell + \gamma_2^\ell L^{\hat{\theta}})L^{\hat{\theta}} + L_2^\ell\gamma^{\hat{\theta}} \triangleq \gamma. \tag{9}$$

With Eq. (8) and Eq. (9), we can conclude $\ell(\lambda, \hat{\theta}(\lambda, S^{tr}), z)$ as a function of $\lambda$ is $L = \mathcal{O}(1)$ Lipschitz continuous and $\gamma = \mathcal{O}(1)$ Lipschitz smooth w.r.t. $K$. By definition, $L$ and $\gamma$ are independent of the training dataset $S^{tr}$, $z$, the random indices $(j_1, j_2, \cdots, j_K)$ and thereby the randomness of $\hat{\theta}$. Thereby, we have $\forall S^{tr} \in Z^n, \forall z \in Z, \forall g \in \mathcal{G}_{\hat{\theta}}, \ell(\lambda, g(\lambda, S^{tr}), z)$ as a function of $\lambda$ is $L = \mathcal{O}(1)$ Lipschitz continuous and $\gamma = \mathcal{O}(1)$ Lipschitz smooth. Similarly, the result also holds for GD. $\quad\square$

# D   UD with GD in the Outer Level

Since GD is deterministic, we can derive a high probability bound for UD with GD in the outer level. Firstly, we define the notion of *uniform stability on validation* for a deterministic HO algorithm.

**Definition 7.** *A deterministic HO algorithm* $\mathbf{A}$ *is* $\beta$-*uniformly stable on validation if for all validation datasets* $S^{val}, S^{'val} \in Z^m$ *such that* $S^{val}, S^{'val}$ *differ in at most one sample, we have*

$$\forall S^{tr} \in Z^{m^{tr}}, \forall z \in Z, \ell(\mathbf{A}(S^{tr}, S^{val}), z) - \ell(\mathbf{A}(S^{tr}, S^{'val}), z) \leq \beta.$$

If a deterministic HO algorithm is $\beta$-uniformly stable on validation, then we have the following high probability bound.

**Theorem 5.** *(Generalization bound of a uniformly stable deterministic algorithm). Suppose a deterministic HO algorithm* $\mathbf{A}$ *is* $\beta$-*uniformly stable on validation,* $S^{tr} \sim (D^{tr})^n$, $S^{val} \sim (D^{val})^m$ *and* $S^{tr}$ *and* $S^{val}$ *are independent, then for all* $\delta \in (0, 1)$, *with probability at least* $1 - \delta$,

$$\ell(\mathbf{A}(S^{tr}, S^{val}), D^{val}) \leq \ell(\mathbf{A}(S^{tr}, S^{val}), S^{val}) + \beta + \sqrt{\frac{(2\beta m + s(\ell))^2 \ln \delta^{-1}}{2m}}.$$

*Proof.* Let $\Phi(S^{tr}, S^{val}) = \ell(\mathbf{A}(S^{tr}, S^{val}), D^{val}) - \ell(\mathbf{A}(S^{tr}, S^{val}), S^{val})$. Suppose $S^{val}, S^{'val} \in Z^m$ differ in at most one point, then

$$|\Phi(S^{tr}, S^{val}) - \Phi(S^{tr}, S^{'val})|$$
$$\leq |\ell(\mathbf{A}(S^{tr}, S^{val}), D^{val}) - \ell(\mathbf{A}(S^{tr}, S^{'val}), D^{val})| + |\ell(\mathbf{A}(S^{tr}, S^{val}), S^{val}) - \ell(\mathbf{A}(S^{tr}, S^{'val}), S^{'val})|.$$

For the first term,

$$|\ell(\mathbf{A}(S^{tr}, S^{val}), D^{val}) - \ell(\mathbf{A}(S^{tr}, S^{'val}), D^{val})|$$
$$= |\mathbf{E}_{z \sim D^{val}} \left[ \ell(\mathbf{A}(S^{tr}, S^{val}), z) - \ell(\mathbf{A}(S^{tr}, S^{'val}), z) \right]| \leq \beta.$$

For the second term,

$$|\ell(\mathbf{A}(S^{tr}, S^{val}), S^{val}) - \ell(\mathbf{A}(S^{tr}, S^{'val}), S^{'val})|$$
$$\leq \frac{1}{m} \sum_{i=1}^{m} |\ell(\mathbf{A}(S^{tr}, S^{val}), z_i^{val}) - \ell(\mathbf{A}(S^{tr}, S^{'val}), z_i^{'val})|$$
$$\leq \frac{s(\ell)}{m} + \frac{m-1}{m}\beta.$$

As a result,

$$|\Phi(S^{tr}, S^{val}) - \Phi(S^{tr}, S^{'val})| \leq \frac{s(\ell)}{m} + 2\beta.$$

According to McDiarmid's inequality, we have for all $\epsilon \in \mathbf{R}^+$,

$$P_{S^{val} \sim (D^{val})^m}(\Phi(S^{tr}, S^{val}) - \mathbf{E}_{S^{val} \sim (D^{val})^m}\left[\Phi(S^{tr}, S^{val})\right] \geq \epsilon) \leq \exp(-2\frac{m\epsilon^2}{(s(\ell) + 2m\beta)^2}).$$

Besides, we have

$$\mathbf{E}_{S^{val} \sim (D^{val})^m}\left[\Phi(S^{tr}, S^{val})\right] = \mathbf{E}_{S^{val} \sim (D^{val})^m}\left[\ell(\mathbf{A}(S^{tr}, S^{val}), D^{val}) - \ell(\mathbf{A}(S^{tr}, S^{val}), S^{val})\right]$$
$$= \mathbf{E}_{S^{val} \sim (D^{val})^m, z \sim D^{val}}\left[\ell(\mathbf{A}(S^{tr}, S^{val}), z) - \ell(\mathbf{A}(S^{tr}, S^{val}), z_1^{val})\right]$$
$$= \mathbf{E}_{S^{val} \sim (D^{val})^m, z \sim D^{val}}\left[\ell(\mathbf{A}(S^{tr}, z, z_2^{val}, \cdots, z_m^{val}), z_1^{val}) - \ell(\mathbf{A}(S^{tr}, S^{val}), z_1^{val})\right] \leq \beta.$$

Thereby, we have for all $\epsilon \in \mathbf{R}^+$,

$$P_{S^{val} \sim (D^{val})^m}(\Phi(S^{tr}, S^{val}) - \beta \geq \epsilon) \leq \exp(-2\frac{m\epsilon^2}{(s(\ell) + 2m\beta)^2}).$$

Notice the above inequality holds for all $S^{tr} \in Z^n$, we further have $\epsilon \in \mathbf{R}^+$,

$$P_{S^{tr} \sim (D^{tr})^n, S^{val} \sim (D^{val})^m}(\Phi(S^{tr}, S^{val}) - \beta \geq \epsilon) \leq \exp(-2\frac{m\epsilon^2}{(s(\ell) + 2m\beta)^2}).$$

Equivalently, we have $\forall \delta \in (0, 1)$,

$$P_{S^{tr} \sim (D^{tr})^n, S^{val} \sim (D^{val})^m}\left(\Phi(S^{tr}, S^{val}) \leq \beta + \sqrt{\frac{(2\beta m + s(\ell))^2 \ln \delta^{-1}}{2m}}\right) \geq 1 - \delta.$$

$\square$

Then we analyze the stability for UD with GD in the outer level. At each iteration in the outer level, it updates the hyperparameter by:

$$\lambda_{t+1} = (1 - \alpha_{t+1}\mu)\lambda_t - \alpha_{t+1}\nabla_\lambda \hat{R}^{val}(\lambda_t, \hat{\theta}(\lambda_t, S^{tr}), S^{val}),$$

where $\alpha_t$ is the learning rate and $\mu$ is the weight decay.

**Theorem 6.** *(Uniform stability of algorithms with GD in the outer level). Suppose $\hat{\theta}$ is a deterministic function and $\forall S^{tr} \in Z^n$, $\forall z \in Z$, $\ell(\lambda, \hat{\theta}(\lambda, S^{tr}), z)$ as a function of $\lambda$ is L-Lipschitz continuous and $\gamma$-Lipschitz smooth. Then, solving Eq. (4) in the full paper with T steps GD, learning rate $\alpha_t \leq \alpha$ and weight decay $\mu \leq \min(\gamma, \frac{1}{\alpha})$ in the outer level is $\beta$-uniformly stable on validation with*

$$\beta = \frac{2L^2}{m(\gamma - \mu)}((1 + \alpha(\gamma - \mu))^T - 1).$$

*Proof.* Suppose $S^{tr} \in Z^n$, we use $F(\lambda, S^{val}, \alpha, \mu) = (1 - \alpha\mu)\lambda - \alpha\nabla_\lambda \hat{R}^{val}(\lambda, \hat{\theta}(\lambda, S^{tr}), S^{val})$ to denote the updating rule of GD, where we omit the dependency on $S^{tr}$ for simplicity. Suppose $S^{val}, S^{'val} \in Z^m$ differ in at most one point, let $\{\lambda_t\}_{t \geq 0}$ and $\{\lambda'_t\}_{t \geq 0}$ be the trace of gradient descent with $S^{val}$ and $S^{'val}$ respectively. Let $\delta_t = ||\lambda_t - \lambda'_t||$, then

$$\begin{aligned} \delta_{t+1} =&||F(\lambda_t, S^{val}, \alpha_{t+1}, \mu) - F(\lambda'_t, S^{'val}, \alpha_{t+1}, \mu)|| \\ \leq&||F(\lambda_t, S^{val}, \alpha_{t+1}, \mu) - F(\lambda'_t, S^{val}, \alpha_{t+1}, \mu)|| + ||F(\lambda'_t, S^{val}, \alpha_{t+1}, \mu) - F(\lambda'_t, S^{'val}, \alpha_{t+1}, \mu)|| \\ \leq&(|1 - \alpha_{t+1}\mu| + \alpha_{t+1}\gamma)\delta_t + \frac{2\alpha_{t+1}L}{m} = (1 + \alpha_{t+1}(\gamma - \mu))\delta_t + \frac{2\alpha_{t+1}L}{m} \\ \leq&(1 + \alpha(\gamma - \mu))\delta_t + \frac{2\alpha L}{m}. \end{aligned}$$

Thereby, we have $\delta_t \leq \frac{2L}{m(\gamma - \mu)}((1 + \alpha(\gamma - \mu))^t - 1)$ for all $t \geq 0$. Finally, we have

$$\forall z \in Z, |\ell(\lambda_T, \hat{\theta}(\lambda_T, S^{tr}), z) - \ell(\lambda'_T, \hat{\theta}(\lambda'_T, S^{tr}), z)| \leq \frac{2L^2}{m(\gamma - \mu)}((1 + \alpha(\gamma - \mu))^T - 1).$$

$\square$

**Remark:** We derive such a bound by using the recursive updates of the outer level GD with the smoothness of the loss function and the inner level optimization. This technique can be directly applied to traditional GD (i.e., GD with one level optimization) to get a stability bound of exponentially increasing w.r.t $T$ and $\mathcal{O}(1/m)$.

# E   Curse of dimensionality in CV

**Lemma 5.** *Suppose $f(\lambda)$, $\lambda \in \Lambda = [0, 1]^d$ is L Lipschitz continuous, $\{\lambda_i\}_{i=1}^T$ are i.i.d. uniform random vectors on $[0, 1]^d$, then $\mathbf{E} \inf_{1 \leq i \leq T} f(\lambda_i) \leq \inf_{\lambda \in \Lambda} f(\lambda) + L\frac{\sqrt{d}}{T^{\frac{1}{d}}}$.*

*Proof.* Let $\lambda^* = \underset{\lambda \in \Lambda}{\arg\min} f(\lambda)$. Firstly, we have $f(\lambda_i) \leq f(\lambda^*) + L||\lambda_i - \lambda^*||$ for all $1 \leq i \leq T$. Thereby,

$$\inf_{1 \leq i \leq T} f(\lambda_i) \leq f(\lambda^*) + L \inf_{1 \leq i \leq T} ||\lambda_i - \lambda^*||.$$

Taking expectation, we have

$$\mathbf{E} \inf_{1 \leq i \leq T} f(\lambda_i) \leq f(\lambda^*) + L\mathbf{E} \inf_{1 \leq i \leq T} ||\lambda_i - \lambda^*||.$$

As for $\mathbf{E} \inf_{1 \leq i \leq T} ||\lambda_i - \lambda^*||$, we have

$$\mathbf{E} \inf_{1 \leq i \leq T} ||\lambda_i - \lambda^*|| = \int_0^\infty P(\inf_{1 \leq i \leq T} ||\lambda_i - \lambda^*|| > t)\mathrm{d}t$$

$$= \int_0^\infty P(||\lambda_1 - \lambda^*|| > t)^T \mathrm{d}t = \int_0^\infty (1 - |B(\lambda^*, t) \cap \Lambda|)^T \mathrm{d}t$$

$$\leq \int_0^\infty (1 - |B(0, t) \cap \Lambda|)^T \mathrm{d}t = \mathbf{E} \inf_{1 \leq i \leq T} ||\lambda_i||$$

$$\leq \mathbf{E} \inf_{1 \leq i \leq T} \sqrt{d} \sup_{1 \leq j \leq d} \lambda_{i,j} = \sqrt{d} \int_0^1 P(\inf_{1 \leq i \leq T} \sup_{1 \leq j \leq d} \lambda_{i,j} > t)\mathrm{d}t$$

$$= \sqrt{d} \int_0^1 P(\sup_{1 \leq j \leq d} \lambda_{1,j} > t)^T \mathrm{d}t = \sqrt{d} \int_0^1 (1 - P(\lambda_{1,1} \leq t)^d)^T \mathrm{d}t$$

$$= \sqrt{d} \int_0^1 (1 - t^d)^T \mathrm{d}t \leq \sqrt{d} \int_0^1 e^{-Tt^d} \mathrm{d}t = \frac{\sqrt{d}}{T^{\frac{1}{d}}} \int_0^{T^{\frac{1}{d}}} e^{-t^d} \mathrm{d}t$$

$$\leq \frac{\sqrt{d}}{T^{\frac{1}{d}}} \int_0^\infty e^{-t^d} \mathrm{d}t = \frac{\sqrt{d}}{T^{\frac{1}{d}}} \int_0^\infty e^{-t} \mathrm{d}t^{\frac{1}{d}} = \frac{\sqrt{d}}{T^{\frac{1}{d}}} \int_0^\infty t^{\frac{1}{d}} e^{-t} \mathrm{d}t = \frac{\sqrt{d}}{T^{\frac{1}{d}}} \Gamma(1 + \frac{1}{d}) \leq \frac{\sqrt{d}}{T^{\frac{1}{d}}}.$$

As a result,

$$\mathbf{E} \inf_{1 \leq i \leq T} f(\lambda_i) \leq f(\lambda^*) + L\frac{\sqrt{d}}{T^{\frac{1}{d}}}.$$

$\square$

The following result implies that CV suffers from curse of dimensionality. CV requires exponentially large $T$ w.r.t. the dimensionality of the $\lambda$ to achieve a reasonably low empirical risk.

**Theorem 7.** *(Curse of dimensionality in CV). Suppose (1) the inner level optimization is solved deterministically, i.e., $\hat{\theta}$ in Eq. (4) in the full paper is a deterministic function, (2) $\{\lambda_t\}_{t=1}^T$ are i.i.d. uniform random vectors taking value in $\Lambda = [0,1]^d$, (3) $\forall S^{tr} \in Z^n$, $\forall z \in Z$, $\ell(\lambda, \hat{\theta}(\lambda, S^{tr}), z)$ as a function of $\lambda$ is $L$ Lipschitz continuous. Let $S^{tr} \sim (D^{tr})^n$ and $S^{val} \sim (D^{val})^m$ be independent, then we have*

$$\mathbf{E}\left[\hat{R}^{val}(\mathbf{A}^{cv}(S^{tr}, S^{val}), S^{val})\right] \leq \mathbf{E}\left[\inf_{\lambda \in \Lambda} \hat{R}^{val}(\lambda, \hat{\theta}(\lambda, S^{tr}), S^{val})\right] + \frac{L\sqrt{d}}{T^{\frac{1}{d}}}.$$

*Proof.* Let $t^*$ be the index of the best hyperparameter, i.e.,

$$t^* = \underset{1 \leq t \leq T}{\arg\min} \hat{R}^{val}(\lambda_t, \hat{\theta}(\lambda_t, S^{tr}), S^{val}),$$

then the output of CV is $\mathbf{A}^{cv}(S^{tr}, S^{val}) = (\lambda_{t^*}, \hat{\theta}(\lambda_{t^*}, S^{tr}))$.

According to Lemma 5, we have

$$\mathbf{E}_{\{\lambda_t\}_{t=1}^T}\left[\hat{R}^{val}(\lambda_{t^*}, \hat{\theta}(\lambda_{t^*}, S^{tr}), S^{val})\right] = \mathbf{E}_{\{\lambda_t\}_{t=1}^T}\left[\inf_{1 \leq t \leq T} \hat{R}^{val}(\lambda_t, \hat{\theta}(\lambda_t, S^{tr}), S^{val})\right]$$

$$\leq \inf_{\lambda \in \Lambda} \hat{R}^{val}(\lambda, \hat{\theta}(\lambda, S^{tr}), S^{val}) + \frac{L\sqrt{d}}{T^{\frac{1}{d}}}.$$

Thereby,

$$\mathbf{E}\left[\hat{R}^{val}(\mathbf{A}^{cv}(S^{tr}, S^{val}), S^{val})\right] = \mathbf{E}_{\{\lambda_t\}_{t=1}^T, S^{tr}, S^{val}}\left[\hat{R}^{val}(\lambda_{t^*}, \hat{\theta}(\lambda_{t^*}, S^{tr}), S^{val})\right]$$

$$\leq \mathbf{E}_{S^{tr}, S^{val}}\left[\inf_{\lambda \in \Lambda} \hat{R}^{val}(\lambda, \hat{\theta}(\lambda, S^{tr}), S^{val})\right] + \frac{L\sqrt{d}}{T^{\frac{1}{d}}}.$$

$\square$

## F  Discussion of the Boundedness Assumption of the Loss Function

The bounded assumption is mild and common (e.g., also used in Theorem 3.12 of [2] and Section 2 in [3]). Indeed, given a machine learning model of a finite number of parameters (e.g. neural networks of finite depth and width used in our experiments), a bounded parameter space (Assumption 1), and a bounded input space (Assumption 1), the feature space is also bounded. Note that previous work makes a similar assumption (at the bottom of Page 9 in [2]) as Assumption 1.

## G  Additional Experiments

### G.1  Generalization Gap

In Figure 1, we plot the generalization gap (estimated by difference between test and validation loss) of UD on the FL and DR experiments. When the $K \geq 64^4$, the generalization gap increases as $K$ increases. These results validate our Theorem 2 and Theorem 3.

In Figure 2, we plot the generalization gap of CV on the FL and DR experiments. There is not a clear relationship between the generalization gap and $K$. These results validate our Theorem 4.

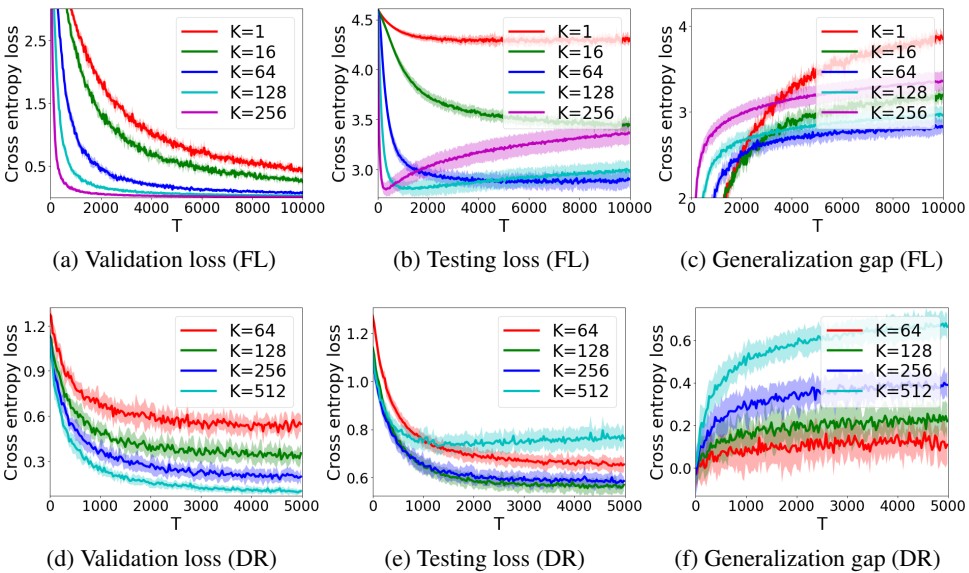

(a) Validation loss (FL)  (b) Testing loss (FL)  (c) Generalization gap (FL)

(d) Validation loss (DR)  (e) Testing loss (DR)  (f) Generalization gap (DR)

Figure 1: The generalization gap of UD in feature learning (FL) and data reweighting (DR).

### G.2  Empirical Verification of the Expectation Bound of CV

We empirically validate the $\mathcal{O}(\sqrt{1/m})$ expectation bound of CV in Theorem 4. In the data reweighting experiment, we chose ten different $m$ from $[10, 1000]$ such that $\sqrt{1/m}$ is distributed linearly and

---

[4]The test loss is dominated by training loss when $K$ is too small due to underfitting on the training dataset.

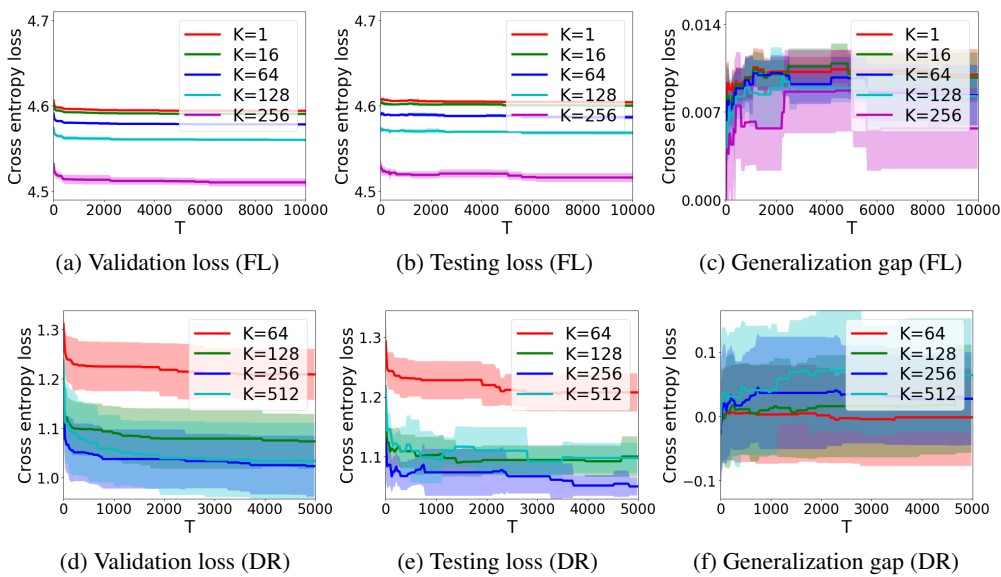

(a) Validation loss (FL)  (b) Testing loss (FL)  (c) Generalization gap (FL)

(d) Validation loss (DR)  (e) Testing loss (DR)  (f) Generalization gap (DR)

Figure 2: The generalization gap of CV in feature learning (FL) and data reweighting (DR).

we plot the curve of the generalization gap v.s. $\sqrt{1/m}$. We fix $T = 1000$ and $K = 64$. We run on 5 different seeds and use the averaged result. As shown in Figure 3, the curve is approximately linear, which accords with our Theorem 4.

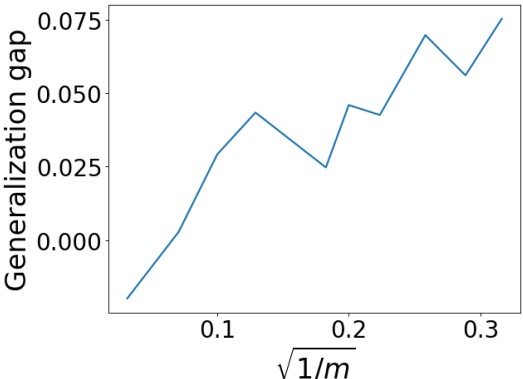

Figure 3: Generalization gap v.s. $\sqrt{1/m}$ of CV in data reweighting (DR).

### G.3 UD with a Smaller Learning Rate in the Inner Level

We also try a smaller learning rate $\eta = 0.1$ in the inner level on the data reweighting task. As shown in Figure 4, it requires $K = 1024$ inner iterations to overfit. This can be explained by our Theorem 3, which implies that a smaller $\eta$ requires a larger $K$ to make the generalization gap unchanged.

### G.4 Experiments with a Smaller Number of Hyperparameters

We also experiment with 4 hyperparameters. We create a two dimensional toy dataset in the feature learning task: $y = x_1^2 + x_2^2 + 0.3\epsilon$, where $x_1, x_2 \sim \text{Uniform}(0, 1)$ and $\epsilon \sim \mathcal{N}(0, 1)$. The number of training data is 10 and the number of validation data is 2. The hyperparameter $\lambda$ is a $2 \times 2$ matrix following the input $x$ and the parameter $\theta$ is a $2 \times 1$ matrix to predict the $y$. The learning rate of the outer level problem is 0.01 and that of the inner level problem is 0.1 and the batch size is 1 in both problems. $K$ is 16 and $T$ is 1000. In this case, the validation losses of UD and CV are comparable

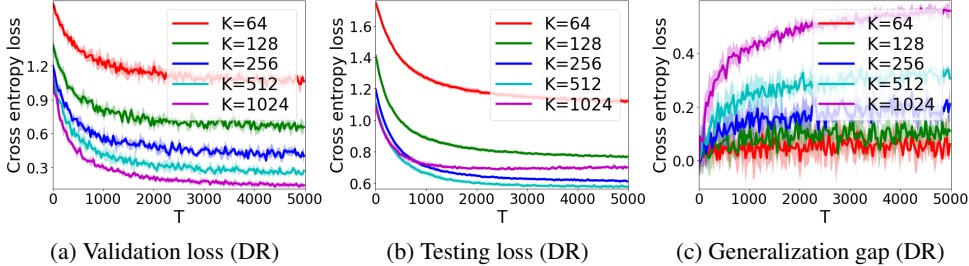

(a) Validation loss (DR)  (b) Testing loss (DR)  (c) Generalization gap (DR)

Figure 4: Results of UD in data reweighting (DR) with a smaller learning rate $\eta = 0.1$. We run on 3 different seeds. The performance of UD is sensitive to the values of $K$ and $T$.

and both algorithms fit well on the validation data. However, UD overfits much severely, leading to a worse testing loss than CV. The results agree with our theory.

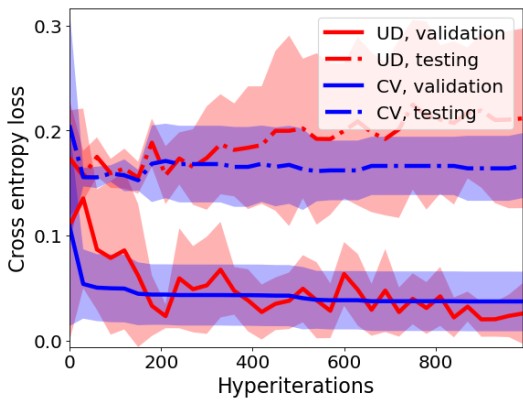

Figure 5: Compare between CV and UD with 4 hyperparameters.