# OpenReview forum: "Stability and Generalization of Bilevel Programming in Hyperparameter Optimization"
_NeurIPS.cc/2021/Conference — NeurIPS 2021 Poster_

### Official Review · Reviewer_AbZm · 2021-06-27

**Rating:** 6
**Confidence:** 4

**Summary:**

In this paper, the authors study bilevel programming for hyperparameter optimization, which involves an outer level optimization on hyperparameter and an inner optimization on hypothesis. Unlike most of existing studies focusing on optimization, the authors study the generalization by using the tool of algorithmic stability. The authors introduce a new stability measure called uniform stability on validation, which considers the sensitivity of the algorithm w.r.t. a change of example in the validation dataset. The authors show the connection between generalization and the proposed algorithmic stability. If the outer level algorithm is SGD, the authors develop uniform stability under some smoothness and Lipschitz assumptions, which were shown to hold under some regularity assumptions on training loss and validation loss. Experimental results are also reported to show the performance of a bilevel programming method (UD) w.r.t different parameters and its comparison with CV.


**Limitations And Societal Impact:**

There seems to be no limitations and societal impact.

**Main Review:**

The application of stability analysis to study generalization bounds of bilevel programming is interesting. The paper is clearly written and easy to follow. The proofs seem to be correct.

My major comment is on the novelty of the analysis. As far as I can see, the arguments in proving Theorem 1 and Theorem 2 follow the existing analysis. In particular, Theorem 2 is proved in the same way as Theorem 3.8 in [12]. The form of stability bounds are also similar and there seems not to be enough novelty in the theoretical analysis. Furthermore, the learning rate there requires to be of the order of $O(1/t)$, which may be very small to get a reasonable low empirical risk. Indeed, it can be shown that SGD with the learning rate $O(1/t)$ requires $O(\exp(1/\epsilon))$ iterations to get the excess empirical accuracy $O(\epsilon)$. This huge number of iterations limits the application of Theorem 2 in practice to understand the performance of Algorithm 1.

In Theorem 3, the authors show the Lipschitz continuity and smoothness of $\ell(\lambda,g(\lambda,S),z)$ under some regularity conditions. However, the Lipschitz & smoothness parameter have an exponential dependency on $K$. Therefore, the learning rate $\eta$ should be very small to get a small Lipschitz & smoothness parameter. Since the authors consider a constant learning rate $\eta$, the learning rate should be of the order of $1/K$ for a moderate Lipschitz & smoothness parameter. This rate may make the algorithm converge slowly in practice.

Algorithm 1 (Line 8) requires to compute $\nabla_{\lambda}\hat{R}^{val}(\lambda,\theta_K^t(\lambda,S^{val}))$. It is not clear to me how to compute this gradient since $\theta_K^t(\lambda,S^{val})$ also depends on $\lambda$. The complexity of computing this gradient may grow as an increasing function of $K$. It would be helpful to explain this gradient computation and its associated complexity.

Minor comments:

There is a superfluous ( in the statement $\gamma=O((())$ in Theorem 3.

Ref [11] and [12] are the same


-------------------------
After authors' response

Thank you for the response. After reading other reviews and response, I would keep my score.

**Time Spent Reviewing:**

7

---

> ### Author Response · Authors · 2021-08-10
> **Response to reviewer AbZm**
>
>
> We thank reviewer AbZm for the positive and constructive comments.
>
> ### Q1: Novelty of the analysis
> We agree that we employ existing techniques. The main contribution of this paper is to formulate a new problem and analyze some practical problems rigorously.
>
> ### Q2: Small learning rate at the outer level
> We admit that we share the same learning rate assumption as in [12], which is a gap between our analysis and the practice. If there is some refined stability analysis of SGD to weaken the assumption, we believe it can be also adopted here to improve our results. We will discuss it in the final version.
>
> ### Q3: Exponential dependence on $K$ and small learning rate at the inner level
> In general, we assume a small learning rate at the inner level. However, as presented in our response to the common concern 2, the stability bound w.r.t. $K$ is $\mathcal{O}(K^2)$ if the inner loss is convex and $\mathcal{O}(1)$ if the inner loss is strongly convex. In these cases, we can use a larger learning rate at the inner level. We will discuss it in the final version.
>
> ### Q4: Compute the gradient $\nabla_\lambda\hat{R}^{val}(\lambda, \theta^t_K(\lambda, S^{val}))$ and the complexity
> As used in many HO problems [8,9,10,23], UD keeps the whole computation graph of $\theta^t_K(\lambda, S^{val})$, which is a composite of SGD updates and is differentiable with respect to $\lambda$. Therefore, the gradient $\nabla_\lambda\hat{R}^{val}(\lambda, \theta^t_K(\lambda, S^{val}))$ can be calculated by the chain rule. The time complexity and memory complexity are both $O(K)$. We will make it clearer in the final version.
>
> ### Q5: Minor comments
> Thanks. We will fix the typos.

---

### Official Review · Reviewer_QRtq · 2021-07-11

**Rating:** 7
**Confidence:** 3

**Summary:**

This paper analyzes, using stability, the generalization properties of a gradient-based bilevel optimization algorithm for hyperparameter optimization.

Bilevel optimization consists in two nested optimization problems: the outer problem aims at minimizing an objective function where some of the variables are a solution of the inner problem. This work considers the setting where inner and outer objectives are the training and validation losses respectively, which arises in hyperparameter optimization. The authors analyze the bilevel optimization algorithm (named UD) consisting in $T$ (stochastic) gradient descent steps on the validation set (consisting of $m$ samples) with gradient computed using unrolling differentiation, i.e. using automatic differentiation to compute the derivative of $K$ steps of (stochastic) gradient descent steps on the train set. This algorithm is used to solve a large variety of machine learning problems like meta-learning or hyperparameter optimization, but its theoretical generalization properties are largely unknown.

This work presents uniform stability bounds which lead to generalization guarantees for stochastic UD without any convexity assumption on the bilevel loss functions. More specifically, it shows a $\tilde O(T^{\mathcal{k}}/m)$ or $\tilde O(f(K))$ (f increasing) upper bound on the absolute value of the expected difference between the true and the empirical risk, where the expectation is taken w.r.t. the algorithm and the training and validation sets.  It further shows a $O(\sqrt{\log(T)/m})$ bound for cross-validation (CV). The analysis shows that with sufficiently low $T$ and $K$ and sufficiently high $m$ UD can be better than CV but also that when $T$ and $K$ are too high UD might suffer more from overfitting on the validation set, a phenomenon which has been observed in practice. The authors also explain how adding L2 regularization in the outer and inner levels can help improve stability. Empirical results on feature learning and data reweighting are in line with the theoretical findings.


**Limitations And Societal Impact:**

The authors discussed part of the limitations in the conclusion section of the paper. Other limitations, which I addressed in the main review, are the exponential dependence on $K$ and the exponential dependence on $T$ for gradient descent.

The authors did not wrote a section addressing societal impact and wrote as a reason that the work is mostly theoretical. I agree with the authors.

**Main Review:**

#### **Originality**
1. The work extends the results of  [12] to the bilevel setting and is one of the first if not the first that studies stability and generalization properties of bilevel problems. The authors could consider [A, B], which study unrolled differentiation (UD) from the optimization point of view with a strongly-convex inner problem. This additional assumption could improve the dependence on the number of inner iterations $K$ in the bound. There are several works concerning bilevel optimization rates with strongly-convex inner problem (see [C] and subsequent works citing it). Maybe the authors could compare their work with some of them.

#### **Significance**
2. Many machine learning problems like meta-learning, hyperparameter optimization, neural architecture search, can be cast into the bilevel framework. Overfitting the validation set can be a serious issue for gradient based bilevel methods and this work is a first step in understanding its theoretical properties. The authors shows interesting generalisation bounds but should discuss more about the exponential dependences on the number of iterations (see 4 and 5 in the quality section).

#### **Quality**
3. The theoretical results seem correct. I checked most of the proofs although I am not very familiar with stability results.

4. I think the authors should explicitly write, in the introduction and later on, that the UD bound (in particular $f$) has an exponential dependence on $K$  and maybe provide the complete UD bound combining theorem 1, 2 and 3 in a corollary or a theorem. I think this exponential dependence should be discussed as a limitation of the approach.

5. The authors present (In the supplementary) a result on the stability of gradient descent  in the non-convex setting which suffers an exponential dependence on the number of steps $T$ and $K$ while enjoying the same $1/m$ dependence as stochastic gradient descent on the number of examples in the validation set $m$. I think it would add value to the paper to bring this result to main part and discuss it more, since in practice the bilevel gradient might be computed using the full or a big part of the validation set (in the experiments, a minibatch is half of the validation set). This is because the validation is usually smaller than the training set and the validation loss is computed $T$ times while the train loss  is computed $T \times K$ times. Also the authors may also discuss how this result contrasts with what stated in [12] at page 26, where the authors could not prove stability results for gradient descent.

6. The experiments show that UD overfits to the validation set with sufficiently high $K$ and $T$, while CV does not suffer from this issue. However, all the experiments have a large number of hyperparameters, hence due to the course of dimensionality, CV does not even achieve a competitive validation accuracy. Experiments with 1 or 2 hyperparameters that show CV in the overfitting regime could be a worthy addition  to the paper. Another valuable addition would be a figure showing the difference between test and validation loss, which could provide a more direct comparison with the theoretical bounds.

#### **Clarity**
7. The main paper and the proofs in the supplementary are quite easy to follow and clearly written.

#### **Final Score**
I am giving a score of 6. I am willing to increase the score if the authors take into account some of the comments in the quality section.

#### **Additional Comments**

1. (1) what does the or mean? Maybe it can be clearer even in the introduction.
2. Line 95: I found it a bit unclear the explanation of differentiable neural architecture search.
3. Line 164: $s(\ell)$ not defined.
4. Line 175 Also assumption 1 is needed.
5. Line 194: reference to existing CV results could be more precise. The high probability bounds for CV start at page 68 of [25].
7.Omniglot setting is a bit unclear (lines 262-267): $lambda$ is the linear layer at the input or output of the network?

#### **Post Rebuttal**
The authors  claimed to make several modifications to the next version of the paper to address most of my concerns. I am thereby increasing the score form 6 to 7.

#### **References**

[12] Moritz Hardt, Ben Recht, and Yoram Singer. Train faster, generalize better: Stability of stochastic gradient descent. In Proceedings of The 33rd International Conference on Machine Learning, pages 1225–1234. PMLR, 2016.

[25] Mehryar Mohri, Afshin Rostamizadeh, and Ameet Talwalkar. Foundations of machine learning. MIT press, 2018.

[A]  R. Grazzi, L. Franceschi, M. Pontil, and S. Salzo, “On the iteration complexity of hypergradient computation,” 2020.

[B]   K. Ji, J. Yang, and Y. Liang, “Bilevel optimization: Nonasymptotic analysis and faster algorithms,” 2020.

[C]  S. Ghadimi and M. Wang, “Approximation methods for bilevel programming,” 2018


**Time Spent Reviewing:**

12

---

> ### Author Response · Authors · 2021-08-10
> **Response to reviewer QRtq**
>
> We thank reviewer QRtq for the positive and constructive comments.
>
> ### Q1: Additional assumption may improve the bound
> We provide improved results when the inner loss is convex or strongly convex. Please see details in our response to the common concern 2. In the final version, we will also discuss the references mentioned in the comment.
>
> ### Q2: Exponential dependence on $K$
> We will present the complete UD bound and discuss that the bound is exponential on $K$ in the final version. Also, we show that this bound is tight in general. Please see details in our response to the common concern 1.
>
> ### Q3: Results of GD
> We will discuss more on GD in the final version. We derive such a bound by using the recursive updates of the outer level GD with the smoothness of the loss function and the inner level optimization. This technique can be directly applied to traditional GD to get a stability bound of exponentially increasing w.r.t. $T$ and $\mathcal{O}(1/m)$. We believe that the authors of [12] (at page 26) meant that they were not able to get a bound better than the exponential dependence on $T$.
>
> ### Q4: Experiments with CV in the overfitting regime
> We experimented with 4 hyperparameters. We created a two dimensional toy dataset in the feature learning task: $y=x_1^2 + x_2^2 + 0.3 \epsilon$, where $x_1, x_2 \sim \mathrm{Uniform}(0, 1)$ and $\epsilon \sim \mathcal{N}(0, 1)$. The number of training data is 10 and the number of validation data is 2. The hyperparameter $\lambda$ is a $2\times 2$ matrix following the input $x$ and the parameter $\theta$ is a $2 \times 1$ matrix to predict the $y$. The learning rate of the outer level problem is 0.01 and that of the inner level problem is 0.1 and the batch size is 1 in both problems. $K$ is 16 and $T$ is 1000. In this case, the validation losses of UD and CV are comparable and both algorithms overfit. However, UD overfits much severely, leading to a worse testing loss than CV. The results agree with our theory.
>
> ### Q5: A figure showing the difference between test and validation loss
> We plotted figures showing the difference between test and validation loss. On the FL and DR experiments, when the $K \geq 64$ (the test loss is dominated by training loss when $K$ is too small due to underfitting on the training dataset), the difference between test and validation loss increases as $K$ increases. These results validate our theory.
> We will add the new results in the final version.
>
> ### Q6: Additional comments
> We will make the text, notation, and reference more precise following your suggestions in the final version.

---

> ### Comment · Reviewer_QRtq · 2021-08-19
> **Updated Review and Score**
>
> Thanks very much for the detailed response. I really appreciate that you are willing to take most of my suggestion into account for the next version of the paper. I updated the review and increased the score.

---

> > ### Author Response · Authors · 2021-08-20
> > **Thanks for the feedback**
> >
> > Thank you very much for the appreciation of our response and the update on the score.

---

### Official Review · Reviewer_eAVT · 2021-07-15

**Rating:** 6
**Confidence:** 3

**Summary:**

This paper is an early theoretical work on generalization of hyperparameter optimization (HO) task. It analyzes the generalization error on validation of unrolled differentiation (UD) and cross validation (CV) methods in expectation and compare them. Besides, a regularized UD algorithm is designed in this paper. Detailed experiments are performed to compare CV with UD, and UD is claimed a better method. Given experimental results show that the regularized UD algorithm improves the performance.

**Limitations And Societal Impact:**

See the above main review

**Main Review:**

Originality: This paper is an early theoretical work on generalization of HO task, giving expectation generalization error bound of UD and CV (previous works always focus on optimization). The differences are clearly demonstrated and related work are cited.

Quality: The submission is technically sound and the claims are well supported. Stability theory is a commonly used method when analyzing generalization error, the method is used appropriately. However, the high probability bound is always better than the expectation bound, this paper does not analyze the high probability one and the reason is not clear. Moreover, the generalization error is applied on validation rather than the testing set, the reason is not clear and the comparison is not given (the explanation given in Section 4.1 is the comparison between training and validation from my perspective).

Clarity: The submission is clearly written and well organized. However, the paper claims that Theorems 2 and 3 give a generalization gap in an order of $O(T^k/m)$ or $O(f(K))$, where $f$ is an increasing function w.r.t $K$. But what kind of $f$ is not explained in the main paper (e.g. $e^K$ and $\sqrt{K}$ are quite different). Moreover, the paper does not explain why the results (both CV and UD) is not related to $n$, the number of training samples, which is an important term from my perspective.

Significance: Via stability theory, the paper gives $O(1/m)$ and $O(\sqrt{1/m})$ generalization gap in expectation for UD and CV, respectively. And the idea can be extended to more related situations. However, previous work gave a high probability $O(\sqrt{1/m})$ generalization gap for CV ([25] as cited in the paper), so it seems the result given in this paper (on CV) is not a better one, or even worse (because we often consider that high probability bounds are better than expectation ones).

**Time Spent Reviewing:**

10hours

---

> ### Author Response · Authors · 2021-08-10
> **Response to reviewer eAVT**
>
> We thank reviewer eAVT for the valuable and constructive comments.
>
> ### Q1: Why using expectation bounds instead of high probability bounds
> Our work aims to explain some mysterious behaviors of UD and CV in practice, and thus we need to compare them under the same theoretical framework to obtain meaningful conclusions. We derive an expectation bound of UD solved by SGD and high probability bounds are nontrivial to obtain due to the randomness in SGD [12] (see details in the common concern 3). Further, we derive an expectation bound of CV instead of using existing high probability ones for a valid comparison with UD.
>
> ### Q2: Generalization is applied on validation instead of test
> This is a potential misunderstanding. In fact, the generalization in this paper is about the test performance (w.r.t. the validation performance).
> In Theorem 1, the notation $\hat{R}^{val}(A(S^{tr}, S^{val}), S^{val})$ refers to the empirical risk on validation as defined in line 81-82. The $R^{val}(A(S^{tr}, S^{val}), D^{val}) $ refers to the expected risk under the test data distribution. We do not distinguish $D^{val} $ from $D^{test}$ in the submission becase they are assumed to be the same distribution as stated in line 78-79. However, we now think it is clearer to denote the expected risk under the test data distribution as $R(A(S^{tr}, S^{val}), D^{test}) $. We will revise the paper to avoid confusion.
>
> ### Q3: Order of $K$ in $f$
> $f(K)$ is $(1+\eta \gamma_\varphi)^{2K}$. Further, we validate the dependence on $K$ is tight (see the common concern 1) and the results can be improved with refined structures of the inner loss (see the common concern 2). We will make it clearer in the final version.
>
> ### Q4: How the bounds are related to the number of training samples $n$
> As presented in lines 149-151, in HO, the distribution of the testing samples is assumed to be the same as that of the validation ones but can differ from that of the training ones. Therefore, **we focus on the generalization gap between the validation and testing data, which does not directly relate to the training data**. The training data has indirect and unknown effects on the bound of UD through the Lipschitz constant $L$. Instead, the number of validation data $m$ explicitly appears in the bounds.
>
> ### Q5: Significance of the expectation bound of CV
> As presented in the response to Q1, we derive an expectation bound of CV instead of using the existing high probability one for a valid comparison with UD. Besides, we present the technical difference between the expectation bound and the existing high probability one and add new experiments to empirically validate the expectation bound of order $\mathcal{O}(\sqrt{1/m})$ (see details in the common concern 3). Furthermore, based on this bound, we obtain meaningful conclusions that CV is hard to overfit (see line 289 and Fig. 2).

---

### Official Review · Reviewer_1Yjf · 2021-07-17

**Rating:** 7
**Confidence:** 4

**Summary:**

In this paper, the authors provide a stability and generalization analysis for bilevel optimization (based on unrolled differentiation, i.e., iterative differentiation via backpropagation) and cross validation (CV) like grid search or random search for hyperparameter optimization. They provide excess uniform stability bounds on validation data  for both UD and CV under certain assumptions, and show that for UD, the stability bound has a scaling of $T^\kappa/m$, where $T$ is the total number of outer iterations and $m$ is the number of validation samples. For CV method, they show that the stability bound scales with $\sqrt{\frac{\log T}{2m}}$. In the appendix, the authors also show that the empirical risk (different from the stability metric) for CV method suffers from a curse of dimensionality. The authors provide experiments showing that UD methods are easier to overfit than CV methods when increasing the number $T,K$ of iteration numbers and a regularization can help to improve the generalization of UD methods.

**Limitations And Societal Impact:**

Yes

**Main Review:**

In this paper, the authors provide a stability and generalization analysis for bilevel optimization (based on unrolled differentiation, i.e., iterative differentiation via backpropagation) and cross validation (CV) like grid search or random search for hyperparameter optimization. They provide excess uniform stability bounds on validation data  for both UD and CV under certain assumptions, and show that for UD, the stability bound has a scaling of $T^\kappa/m$, where $T$ is the total number of outer iterations and $m$ is the number of validation samples. For CV method, they show that the stability bound scales with $\sqrt{\frac{\log T}{2m}}$. In the appendix, the authors also show that the empirical risk (different from the stability metric) for CV method suffers from a curse of dimensionality. The authors provide experiments showing that UD methods are easier to overfit than CV methods when increasing the number $T,K$ of iteration numbers and a regularization can help to improve the generalization of UD methods. My detailed comments are given as below.

Pros:
1.	Bilevel optimization and hyperparameter optimization are both  important and timely topics recently, and this paper provides some theoretical investigation on the stability and generalization in these two areas.
2.	The comparison between UD (i.e., gradient-based bilevel optimization methods) and CV (such as random search methods) is a good step to help people understand their differences.
3.	The authors provide a theoretical justification on the tradeoff on $K$ and $T$ (i.e., the overfitting and underfitting for different choices of $K,T$)  found by Franceschi et al., 2018.
4.	The experiments support the theoretical results to certain degree.

However, I have several concerns about this submission.

Cons:
1.	My first concern is the tightness of the results. For UD methods, it seems to that the stability bounds increase exponentially with the number $K$ of the inner gradient steps, and such a large dependence is not verified via lower bounds or experiments. It seems to me the authors derive such a dependence by using the recursive updates of gradient descent combined with smoothness of the inner loss functions (this is why the base number is $1+\eta\gamma_{\varphi}$ in the worst case). However, for the case when the inner loss has refined structure, i.e., convexity or strong convexity, I guess the bound may not increase with $K$ (if this is true, the claims on the tradeoff on $K$ should be rephrased depending on the geometry of the inner loop). Therefore, I first hope there are some comments or justification on the tightness of such exponential dependences and also wish the authors can check whether such dependences still hold for convex or strongly-convex inner loop (which are also interesting and common geometries for bilevel optimization).
2.	My second concern is that the assumptions made in this paper are a little bit restrictive. For example, they require the objective function value to be bounded, which is a strong assumption given unbounded feature space (e.g., this is hard to hold even for linear classifier). In addition, they require the loss function is second- and third-order continuously defendable, and hence such assumptions will eliminate the widely used ReLU activation (indeed the authors motivate their assumptions using neural networks with smooth operators).
3.	This paper is poorly structured and not very easy to follow. For example, in Theorem 2 and Theorem 3, it is not clear to me what is $\mathcal{G}_{\hat\theta}$ and what is $g(\lambda,S^{tr})$. In addition, the assumptions are made on both $l$ and $\varphi_i$, and it takes me some time to figure out $l$ is a loss over validation data and $\varphi_i $ is a loss on the $i^{th}$ training data sample. Why do not use $l_i$ for the $i^{th}$ validation data for a consistence?  I may miss something, but it takes me much time to search for the definitions of such notations. I suggest that the authors recall the definitions of them before the statements of the results.
4.	For the results on CV, Mohri et al., 2018 derived the high-probability bounds, whereas this paper provides a bound in expectation. Based on my experience, high-probability bounds are often more harder than expectation bounds. The authors state that their expectation bounds involve new developments of independent interest. For this reason, I highly suggest that the authors can sketch the novelty of their bounds, and how their bounds differ from the existing high probability ones. This will strengthen this paper a lot.
5.	For the experiments, I have one question. The authors set a learning rate 0.3 for the inner loop for UD method, which may be a little bit large. A large learning rate may prevent the inner iterates from approaching the optimal point. Therefore, I suggest that the authors choose a smaller learning rate, e.g., 0.1, 0.05, or 0.01 to verify this phenomenon.

In summary, I am not very excited about the results in this paper, because the findings are not new and the techniques such as the uniform stability are a little bit standard. In addition, the results and presentation need more time to be enhanced or revised to reach the bar of NeurIPS. If the authors can provide a good response, especially the tightness check via any lower bound, I am open to increase my score.


**Time Spent Reviewing:**

6

---

> ### Author Response · Authors · 2021-08-10
> **Response to reviewer 1Yjf**
>
> We thank reviewer 1Yjf for the insightful and constructive comments.
>
> ### Q1: Tightness of the results and improvements with refined structures of the inner loss
> We show the results are tight in general and present a better result with refined structures of the inner loss. Please see details in our response to the common concerns 1&2 respectively.
>
> ### Q2: Assumptions of bounded and continuously differentiable loss functions
> * The bounded assumption is mild and common (e.g., also used in Theorem 3.12 of [12] and Section 2 in [1*]). Indeed, given a machine learning model of a finite number of parameters (e.g. neural networks of finite depth and width used in our experiments), a bounded parameter space (Assumption 1), and a bounded input space (Assumption 1), the feature space is also bounded. Note that previous work makes a similar assumption (at the bottom of Page 9 in [12]) as Assumption 1.
>
> * The continuously differentiable assumption does not hold for ReLU. However, we argue that there are many smooth approximations of ReLU including Softplus, Gelu [2*], and Lipswish[3*], which satisfy the assumption and achieve promising results in classification and deep generative modeling.
>
> We will add the above discussion in the final version.
>
> ### Q3: Writing and notation
> Thanks. We will use consistent notations and add a separated section to present all notations.
>
> ### Q4: Novelty of the expectation bound
> We discuss the technical difference between the expectation bound and the high probability bound, and verify the expectation bound empirically. See details in common concern 3.
>
> ### Q5: Smaller learning rate in the inner loop
> Thanks for the suggestion. We tried smaller learning rates and got a similar overfitting phenomenon of UD as shown in Fig.1 (d) of the paper. In such a case, we used a larger $K$. For instance, a learning rate of $\eta=0.1$ requires $K=1024$ inner iterations to overfit. This can be explained by our Theorem 3, which implies that a smaller $\eta$ requires a larger $K$ to make the generalization gap unchanged. We will add the new results in the final version.
>
> [1*] Shalev-Shwartz S, Shamir O, Srebro N, et al. Learnability, stability, and uniform convergence. The Journal of Machine Learning Research, 2010, 11: 2635-2670.
>
> [2*] Hendrycks D, Gimpel K. Gaussian error linear units (gelus). arXiv preprint arXiv:1606.08415, 2016.
>
> [3*] Chen R T Q, Behrmann J, Duvenaud D, et al. Residual flows for invertible generative modeling. arXiv preprint arXiv:1906.02735, 2019.

---

> > ### Comment · Reviewer_1Yjf · 2021-08-28
> > **Thanks for the updates**
> >
> > Dear authors,
> >
> > Thanks for your detailed response. I am satisfied with the worst-case constructions of the exponential Lipschitz constant L. It would be better if the experiments with smaller learning rate can be added in the revision. I am also hoping the following related works should be added in the revision, which all characterize the convergence rate of bilevel optimization algorithms.
> >
> > Convergence of bilevel optimization:
> >
> > 1. Ghadimi, Saeed, and Mengdi Wang. "Approximation methods for bilevel programming." arXiv preprint arXiv:1802.02246 (2018).
> > 2. Ji, Kaiyi, Junjie Yang, and Yingbin Liang. "Bilevel optimization: Convergence analysis and enhanced design." International Conference on Machine Learning. PMLR, 2021.
> > 3. Ji, Kaiyi, and Yingbin Liang. "Lower Bounds and Accelerated Algorithms for Bilevel Optimization." arXiv preprint arXiv:2102.03926 (2021).
> > 4........etc
> >
> > Bilevel optimization for meta-learning:
> >
> > 1. Ji, K., Lee, J. D., Liang, Y., & Poor, H. V. (2020). Convergence of meta-learning with task-specific adaptation over partial parameters. NeurIPS 2020.
> > 2. ....etc
> >
> > I have increased my score from 5 to 6 due to the some missing bilevel works in the related work section. I will further increase my score to 7 if more related works are properly added.
> >
> > Best, reviewer

---

> > > ### Author Response · Authors · 2021-08-28
> > > **Thanks for the update and valuable feedback**
> > >
> > > Dear Reviewer 1Yjf,
> > >
> > > Thank you very much for the update of rating as well as the valuable feedback on related work. We are happy to know that you found the response satisfactory. We will definitely like to include a more thorough discussion on the related work as well as the results with smaller learning rate in the final version. We hope you might view this as sufficient reason to further raise your score.
> > >
> > > Best,
> > > Authors

---

> > > > ### Comment · Reviewer_1Yjf · 2021-08-28
> > > > **Thanks for the response**
> > > >
> > > > Dear Authors,
> > > >
> > > > Thanks for the quick response and confirmation, and I have updated my score to 7.
> > > >
> > > > Best,
> > > > Reviewer

---

> > > > > ### Author Response · Authors · 2021-08-29
> > > > > **Thanks for the update!**
> > > > >
> > > > > Thank you very much for the update. We highly appreciate it.

---

### Official Review · Reviewer_ZjhP · 2021-07-26

**Rating:** 5
**Confidence:** 2

**Summary:**

The paper studies bilevel optimization problems, more precisely hyperparameter optimization (HO).
It aims at understanding the generalization behavior of algorithms used to solve the latter.
In HO, a validation loss (typically prediction mean squared error on left out data) is minimized with respect to some hyperparameters (eg, regularization strength)
subject to the constraint that the parameter/hypothesis (eg the regression coefficient) minimize an inner loss (depending on the hyperparameter) on a training data.



**Limitations And Societal Impact:**

.

**Main Review:**

Cross validation coupled with grid search is a classical procedure to solve HO, but scales badly with the number of parameters to tune.
Recently the focus has shifted to 1st order, gradient based methods, based either on implicit or iterative differentiation in the inner problem to compute the gradient of the outer loss with respect to the hyperparameter.

Some bilevel algorithms overfit to the validation set (even if it is never seen in the inner problem).
To understand this, the paper presents a notion of uniform stability on the validation set when HO is solved with stochastic gradient descent.
The paper also proposes to regularize both the inner and outer optimization problems to avoid overfitting of 1st order bilevel algorithms.

Clarifications :
- it seems to me that there is a confusion between cross validation (a formulation of the bilvel optimization problem where the outer cost is the average over folds of performance on left out data) and grid search (applicable to any HO problem, where the hyperparameter space is discretized). The authors tend to use CV when they mean grid search, this should be fixed (since, for example, CV can be solved with 1st order methods)
- how is the Lipschitzness assumption on the Loss in thm 2 valid ? For a simple ridge regression, as a function of lambda, can it be lipschitz without bounding the coefficients ? the strong convexity assumption also seems unverified in practice. Can you exemplify some losses which enter into your framework ?

**Time Spent Reviewing:**

3

---

> ### Author Response · Authors · 2021-08-10
> **Response to reviewer ZjhP**
>
> We thank reviewer ZjhP for the valuable and constructive comments.
>
> ### Q1: Confusion between "cross-validation" and "grid search"
> Thanks for the suggestion. In the final version, we will use "CV-GS" instead of "CV" when referring to the baseline method.
>
> ### Q2: Lipschitzness assumption on the loss in Theorem 2
> The Lipschitzness assumption in Theorem 2 is relatively mild. We provide a sufficient condition (e.g., Assumption 1-4) in Theorem 3 to ensure the validness of the Lipschitzness assumption. Previous work makes a similar assumption (at the bottom of Page 9 in [12]) as Assumption 1, *which requires the parameter space to be bounded*. Assumptions 2&3&4 can be satisfied by using smooth loss (e.g., squared loss, cross-entropy, and many others) and smooth activations (e.g. sigmoid, tanh, softplus, and many others) if neural networks are considered. Thus, our framework can be applied to all of the examples used in our experiments (e.g. feature learning and data reweighting).
>
> As for the ridge regression example mentioned in the comment, if we do not assume the coefficients are bounded, then it violates Assumption 1. However, we argue that if a proper hyperparameter is set, the optimization trajectory of the ridge regression usually sticks in a bounded space due to the presence of the regularization and therefore Assumption 1 holds.
>
> Further, we clarify that we **do not assume the strong convexity** in our main results. If the assumption does hold, we can obtain better results. Please see our response to the common concern 2.
>
> We'll make the above discussion more explicit in the final version.

---

> > ### Author Response · Authors · 2021-08-30
> > **Looking forward to further feedback**
> >
> > Dear Reviewer ZjhP,
> >
> > Thank you again for the great efforts and the valuable comments. We have carefully addressed the main concerns in detail. We hope you might find the response satisfactory (similar as the other reviewers). As the discussion phase is about to close, we are very much looking forward to hearing from you about any further feedback. We will be very happy to clarify further concerns (if any).
> >
> > Best,
> > Authors

---

### Author Response · Authors · 2021-08-10
**Common concerns from reviewers**

We thank all reviewers for their valuable and constructive comments. We address the common concerns here and post a point-to-point response to each reviewer as well. We believe the quality of the paper has been improved following the reviewers' suggestions.

### Common concern 1 (from reviewer 1Yjf, QRtq): Tightness of the exponential dependence on $K$
We show that the exponential dependence on $K$ is tight in general. Indeed, we construct a worst case where the Lipschitz constant $L$ in Theorem 3 increases at least exponentially w.r.t. $K$. It is a feature learning example with a small neural network. The model has one parameter $\theta$ and one hyperparameter $\lambda$ and uses squared activation function [1*]. We use the squared loss. The data distribution is any distribution in the support $Z=\\{(x, y) : \frac{1}{2} \le x \le 1, 1 \le y \le 2\\}$. The parameter space and hyperparameter space are $\Theta = [0, 1]$ and $\Lambda = [0, \frac{1}{4}]$ respectively. Formally, the loss function is $ \ell(\lambda, \theta, z) = (y-\lambda(\theta x)^2)^2$. The inner loop is solved by SGD with a learning rate $\eta$.

Note that this problem is not convex w.r.t. $\theta$. We can recursively construct a lower bound of $L$ and finally obtain $L = \Omega((1+\frac{3}{16} \eta)^K)$. According to Theorem 2, the stability bound also increases exponentially w.r.t. $K$ in this worst case. Therefore, the exponential dependence on $K$ is tight in general. We will add the new results in the final version.

### Common concern 2 (from reviewer 1Yjf, QRtq, AbZm): Improvement under stronger assumptions
Indeed, assuming that the inner loss is convex or strongly convex, the bounds become tighter. The dependence on $K$ of the stability bound is $\mathcal{O}(K^2)$ in the convex case and $\mathcal{O}(1)$ in a strongly convex case. They can be summarized in the following theorems.
* When the inner loss $\varphi_j(\lambda, \theta, z)$ as a function $\theta$ is convex for every $j, \lambda, z$ and $\eta \leq 2/\gamma_\varphi$, we have $\beta = \mathcal{O}(K^2)$.
* When $\varphi_j(\lambda, \theta, z)$ as a function $\theta$ is $\tau$-strongly convex for every $j, \lambda, z$ and $\eta \leq 1/\gamma_\varphi$, we have $\beta = \mathcal{O}(1)$ w.r.t. $K$.

The proofs of the new results are similar to Theorem 3.8 and Theorem 3.9 in [12], respectively. We will add the new results and proofs in the final version.

### Common concern 3 (from reviewer 1Yjf, eAVT): Issues on expectation bounds
We clarify the motivation of using expectation bounds to compare UD and CV, present the technical difference between the CV bound and existing high probability ones and add an experiment to verify the CV bound empirically.

* **Motivation**: Our work aims to explain some mysterious behaviors of UD and CV in practice, and thus we need to compare them under the same theoretical framework to obtain meaningful conclusions. We derive an expectation bound of UD solved by SGD as in [12]. In such cases, *“technically, high probability bounds are hard to obtain by the fact that SGD is itself randomized, and thus a concentration inequality must be devised to account for both the randomness in the data and the training algorithm”* [12]. Further, we derive the expectation bound of CV instead of using existing high probability ones for a valid comparison with UD.
* **Technical difference between the expectation bound of CV and existing high probability ones**: Technically, the expectation bound of CV is proved via the property of the maximum of a set of subgaussian random variables (see Theorem 1.14 in [2*]), which is distinct from the union bound used in the high probability bound of CV [25]. We will discuss them in detail in the final version.
* **Verification of the CV bound**: We empirically validate the $\mathcal{O}(\sqrt{1/m})$ expectation bound of CV in Theorem 4. In the data reweighting experiment, we chose ten different $m$ from $[10, 1000]$ such that $\sqrt{1/m}$ is distributed linearly and we plot the curve of the generalization gap v.s. $\sqrt{1/m}$. The curve is approximately linear, which accords with our Theorem 4. We will add the results in the final version.

Based on the expectation bounds, we obtain meaningful conclusions that UD may suffer from overfitting on validation (see line 283-285 and Fig. 1) and CV is hard to overfit (see line 289 and Fig. 2). We will add the discussion in the final version.

[1*] Ali R E, So J, Avestimehr A S. On polynomial approximations for privacy-preserving and verifiable relu networks[J]. arXiv preprint arXiv:2011.05530, 2020.

[2*] Rigollet P. 18. s997: High dimensional statistics[J]. Lecture Notes), Cambridge, MA, USA: MIT Open-CourseWare, 2015.

---

### Author Response · Authors · 2021-08-28
**Looking forward to further feedback**

Dear AC and Reviewers,

Thank you again for the great efforts and the valuable comments. We have carefully addressed the main concerns in detail. We hope you might find the response satisfactory. As the discussion phase is about to close, we are very much looking forward to hearing from you about any further feedback. We will be very happy to clarify any further concerns (if any).

Best,
Authors

---

### Decision · Program_Chairs · 2021-09-27

**Decision:**

Accept (Poster)

**Comment:**

The paper  is among the early theoretical work on studying the stability and generalization of hyperparameter optimization (HO) task in contrast to the optimization properties widely studied in the literature.  It used a notion of uniform stability for validation to get the rate of $O({T^\kappa\over m})$ in comparison to $O({\sqrt{\log T}\over m})$ using CV approaches. There are various interesting discussions on the implication of the comparisons on the possible superior performance of UD in HO framework. Experiments validate the theoretical findings. The reviewers' concerns/questions on the tightness of the exponential dependence on K , possible improvement of the bounds if the inner loss is convex or strongly, and other questions on the experiments are well responded in the rebuttal.